# Stabilizing Reinforcement Learning for Diffusion Language Models

**Jianyuan Zhong** [* 1 2]  **Kaibo Wang** [* 1 3]  **Ding Ding** [* 1 3]  **Zijin Feng** [1]  **Haoli Bai** [1]  **Yang Xiang** [3]  **Jiacheng Sun** [1]
**Qiang Xu** [2 4]

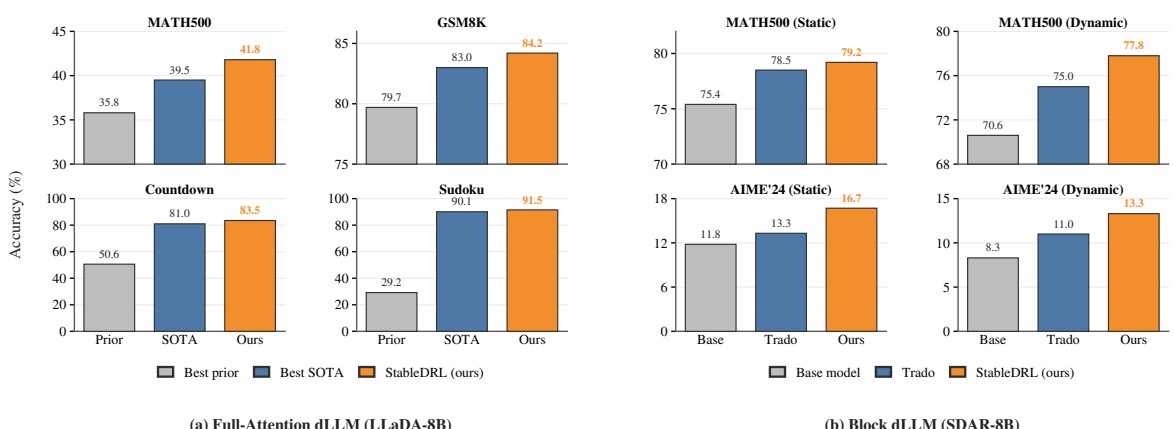

(a) Full-Attention dLLM (LLaDA-8B)   (b) Block dLLM (SDAR-8B)

*Figure 1.* **StableDRL enables stable full-parameter RL training on full-attention and block dLLMs.** The left panel reports full-attention dLLM results using LLaDA-8B (Nie et al., 2025b); the right panel reports block diffusion results using SDAR-8B (Cheng et al., 2025). Best Prior denotes WD1 (Tang et al., 2025); Best SOTA denotes the stronger result between ESPO and SPG (Ou et al., 2025a; Wang et al., 2025a) for each task.

## Abstract

Group Relative Policy Optimization (GRPO) is highly effective for post-training autoregressive language models, yet its direct application to diffusion large language models (dLLMs) often triggers reward collapse. We identify two sources of incompatibility: dLLM sequence likelihoods are intractable and must be estimated through noisy likelihood proxies, and standard GRPO's conditional clipping and fixed group-size normalization are not designed for such estimated ratios. We show that these effects can form a self-reinforcing instability loop in which noisy ratio outliers induce gradient spikes and policy drift, further increasing future ratio variance. We propose StableDRL, a reformulation of GRPO for dLLMs that applies unconditional upper clip-

ping to suppress outlier-induced spikes and self-normalization to constrain updates within the convex hull of per-sample advantage-weighted update directions, and we extend it to block-wise diffusion via staircase attention. On full-attention and block diffusion language models, Stable-DRL enables stable full-parameter reinforcement learning and improves reasoning performance over prior dLLM RL baselines on MATH500, GSM8K, Countdown, Sudoku, and AIME. Source code and experimental configurations are available at https://github.com/cure-lab/StableDRL.git.

## 1 Introduction

Discrete Diffusion Large Language Models (dLLMs) have emerged as a compelling alternative to autoregressive (AR) models, intrinsically supporting parallel decoding and bidirectional context modeling (Sahoo et al., 2024; Nie et al., 2025b; Wu et al., 2025; Yang et al., 2025b). While Group Relative Policy Optimization (GRPO) has proven highly effective for reinforcement learning (RL) in the AR paradigm, its direct application to dLLMs leads to severe instability. As shown in Figure 2(a), full-parameter GRPO training on dLLMs exhibits an abrupt reward collapse at ∼ 300 steps.

[*]Equal contribution  [1]Huawei Foundation Model Department [2]The Chinese University of Hong Kong, Hong Kong SAR, China [3]The Hong Kong University of Science and Technology, Hong Kong SAR, China [4]Shenzhen Loop Area Institute. Correspondence to: Zijin Feng <zijin.feng@huawei.com>, Jiacheng Sun <sunjiacheng1@huawei.com>, Qiang Xu <qxu@cse.cuhk.edu.hk>.

*Proceedings of the 43rd International Conference on Machine Learning*, Seoul, South Korea. PMLR 306, 2026. Copyright 2026 by the author(s).

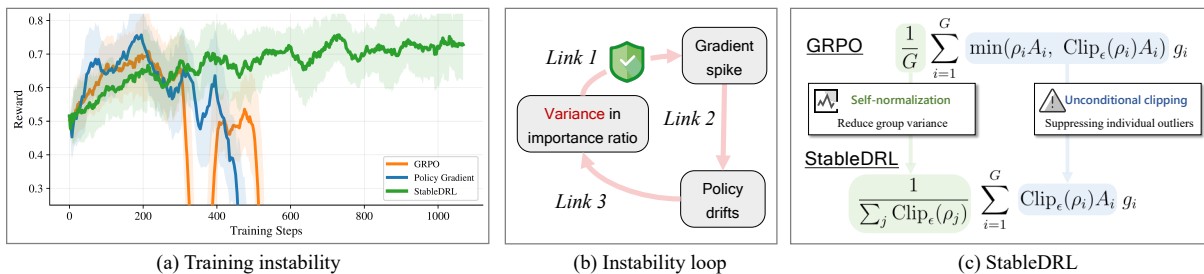

*Figure 2.* **(a) Training instability.** Full-parameter dLLM RL is fragile under direct GRPO-style training with noisy estimated importance ratios. **(b) Instability loop.** Our analysis focuses on the additional spike-drift instability caused by noisy estimated ratios: estimation noise triggers gradient spikes and policy drift, creating a self-reinforcing cycle that amplifies the variance of future importance ratios. **(c) StableDRL.** To address this, we propose a reformulated GRPO for noisy importance ratios. By employing unconditional upper clipping and self-normalization, StableDRL effectively breaks the instability loop.

The incompatibility between GRPO and dLLMs stems from two factors: (i) the intractability of importance ratios in dLLMs (Ou et al., 2025c;a) and (ii) GRPO's lack of adaptation to estimated importance ratios (Section 3.1). GRPO updates a target policy using data sampled from a behavior policy based on the importance ratios, defined as the ratio of their sequence probabilities. While this probability is tractable for AR models, it is intractable for dLLMs and must be estimated via sampling. Prior research has focused on the dLLM aspect, refining importance ratio estimation using mean-field approximations (Zhao et al., 2025b; Tang et al., 2025) or Evidence Lower Bound (ELBO) estimations (Yang et al., 2025b; Wang et al., 2025a; Ou et al., 2025a). Although these approaches yield performance gains, they empirically remain prone to training instability.

We attribute the instability in dLLMs to two design flaws in standard GRPO, which is inherently sensitive to the noisy importance ratios. First, the clipping mechanism in GRPO is conditional. In AR models, this mechanism accelerates the policy's return to the trust region. In dLLMs, however, model-agnostic estimation noise allows the clipping condition to be anomalously bypassed, triggering gradient spikes. Second, GRPO normalizes updates by the fixed group size. Given the high variance of importance ratio estimation in dLLMs, this static normalization results in drastic fluctuations in gradient magnitude, destabilizing the optimization process. To address the instability, we first analyze the underlying mechanism and then propose a stable GRPO variant tailored for dLLMs.

We theoretically and empirically demonstrate how these flaws precipitate a self-reinforcing *instability loop*, leading to reward collapse in the full-parameter dLLM RL settings we study. As shown in Figure 2(b), noisy importance ratios first induce gradient spikes under the GRPO update (Link 1). These spikes degrade the target policy, causing it to deviate significantly from the behavior policy (Link 2). This deviation, in turn, exacerbates the variance of importance ratios in subsequent steps (Link 3). Under the stated tail and local smoothness assumptions, our analysis shows that once drift increases after a spike, the lower bound on future spike probability is non-decreasing, formalizing a self-reinforcing failure mechanism.

To stabilize training, we propose **StableDRL** to break the instability loop at its source (Link 1). As illustrated in Figure 2(c), StableDRL incorporates two components. (i) We introduce *unconditional clipping*, which enforces an unconditional upper bound on importance ratios regardless of the advantage. This prevents outliers from generating gradient spikes. (ii) We employ *self-normalization*. Instead of dividing by the group size, we normalize the update by the sum of clipped importance ratios. This constrains the update within the convex hull of per-sample advantage-weighted update directions. Furthermore, we extend StableDRL to block diffusion models (Cheng et al., 2025) via a *staircase attention* mechanism, enabling leakage-free probability estimation.

To the best of our knowledge, StableDRL is the first method to enable stable, full-parameter RL training on both full-attention and block dLLMs for over 1,000 steps. This sustained stability effectively increases the volume of valid training rollout samples, allowing the model to fully unlock its reasoning capabilities and empirically achieve state-of-the-art performance in dLLM reasoning tasks. Our contributions are summarized as follows:

- We theoretically and empirically identify the self-reinforcing instability loop that causes reward collapse when GRPO is applied to dLLMs.

- We propose StableDRL, a novel reinforcement learning framework for stabilizing the full-parameter training of dLLMs through unconditional clipping and self-normalization.

- Comprehensive experiments validate the effectiveness of our StableDRL on both full-attention and block dLLMs, showing higher training stability and significant accuracy gain over prior best-in-class methods.

## 2  Background

### 2.1  Masked Diffusion Language Models

Masked diffusion language models (MDLMs) (Nie et al., 2025b; Sahoo et al., 2024) formulate text generation as a discrete diffusion process modeled by a continuous-time Markov chain. Given a clean sequence $x_0 \in \mathcal{V}^n$, the forward process $q(x_t|x_0)$ independently corrupts tokens by transitioning them to a special mask token M according to a schedule $t \in [0, 1]$. The generative process reverses this corruption by learning a denoiser $\pi_\theta(x_0|x_t)$ to reconstruct the original data from the latent state.

Unlike AR models where the exact sequence log-likelihood $\log \pi_\theta(x_0)$ is computationally tractable via the chain rule, that of MDLMs is intractable, as it requires marginalizing over all $n!$ masking trajectories (Ou et al., 2025c). Consequently, training optimizes the Evidence Lower Bound (ELBO) (Wu et al., 2025; Ou et al., 2025b), denoted as $\mathcal{L}_\theta(x_0)$ with $\mathcal{L}_\theta(x_0) \leq \log \pi_\theta(x_0)$:

$$\mathcal{L}_\theta(x_0) = \mathbb{E}_{t,x_t}\left[\frac{1}{t}\sum_{i=1}^n \mathbb{1}(x_t^i = \text{M}) \log \pi_\theta(x_0^i|x_t)\right]. \quad (1)$$

In practice, this expectation is approximated via Monte Carlo (MC) sampling. We denote the single-sample MC estimator of the ELBO as $\hat{\mathcal{L}}_\theta(x_0)$.

### 2.2  Reinforcement Learning with dLLMs

We focus on fine-tuning dLLMs to maximize a reward function $R(x)$ using policy gradient methods. The standard objective is to maximize the expected return $\mathcal{J}(\theta) = \mathbb{E}_{x \sim \pi_\theta}[R(x)]$. Modern on-policy algorithms, such as PPO (Schulman et al., 2017) and GRPO (Shao et al., 2024), improve sample efficiency by utilizing importance sampling to update the policy using trajectories collected from a behavior policy $\pi_{\theta_{\text{old}}}$.

**Group Relative Policy Optimization (GRPO).** GRPO eliminates the value function critic by estimating the baseline using the group average of rewards. For a group of rollouts $\{x_1, \ldots, x_G\}$ sampled from $\pi_{\theta_{\text{old}}}$ conditioned on prompt $c$, the gradient update formula is:

$$\nabla_\theta \mathcal{J}_{\text{GRPO}} = \mathbb{E}\left[\frac{1}{G}\sum_{j=1}^G \min\left(\rho_j A_j, \text{clip}_\epsilon(\rho_j)A_j\right)g_j\right], \quad (2)$$

where $A_j$ is the advantage standardized within the group, $\rho_j(x) = \frac{\pi_\theta(x)}{\pi_{\theta_{\text{old}}}(x)}$ is the importance ratio, and clip$(\cdot)$ ensure a trused region of $[1 - \epsilon, 1 + \epsilon]$. For simplicity, We omit dependencies on $x, x \sim \pi_{old}$ and denote the gradient $\nabla_\theta \log \pi_\theta(x)$ by $g$. GRPO retains the min-clip operation from PPO, implicitly regularizing the divergence between the target policy and the behavior policy. When $\rho$ tends to move away from the trust region (e.g., $\rho > 1 + \epsilon, A > 0$),

the clip operation limits the update step size. When it tends to return to the trust region ($\rho > 1 + \epsilon, A < 0$), employing unclipped step sizes to accelerate training.

**Challenges in adapting GRPO to dLLMs.** Prior works primarily improve importance ratio estimation methods and directly porting them to Eq. (2). While earlier methods like D1 and WD1 (Zhao et al., 2025b; Tang et al., 2025) attempted a one-step mean-field approximation, Zhao et al. (2025a) showed this to be inaccurate. As a result, current state-of-the-art approaches (Yang et al., 2025b; Wang et al., 2025a; Ou et al., 2025a) use multi-step Monte Carlo sampling to estimate likelihood via the Evidence Lower Bound (ELBO). However, in practical on-policy RL with limited MC steps ($m \leq 5$) (Ou et al., 2025a; Wang et al., 2025a), the estimation suffers from noise and outliers. In Sec. 3.1, we show that the combination of this estimation noise and the standard GRPO formulation causes training instability.

## 3  Methodology

In this section, we first diagnose the root cause of the observed reward collapse in GRPO, identifying an *instability loop* driven by the long-tail noise of importance ratios (Sec. 3.1). We then propose **StableDRL**, which mitigates this instability through unconditional clipping and self-normalization (Sec. 3.2). Finally, we provide a theoretical justification for our method (Sec. 3.3).

### 3.1  Understanding Instability in dLLM RL Training

As current state-of-the-arts utilize Monte Carlo sampling to estimate the intractable importance ratios of dLLMs, we model the training instability as a three-stage process: (i) noise in estimated importance ratios forms a long-tail distribution; (ii) high variance and outliers generate gradient spikes; and (iii) these spikes induce policy drift, which amplifies future variances of the estimated importance ratios, closing the instability loop.

**(i) Variance in importance ratios.** Let $\eta(x)$ denote noise from the estimation error of the ELBO, such that $\hat{\mathcal{L}}_\theta(x) = \mathcal{L}_\theta(x) + \eta(x)$. The estimated $\hat{\rho}(x)$ can be decomposed into a policy drift term and a noise term:

$$\hat{\rho}(x) = \frac{\exp \hat{\mathcal{L}}_\theta(x)}{\exp \hat{\mathcal{L}}_{\theta_{\text{old}}}(x)} = \underbrace{\exp(\Delta\mathcal{L}(x))}_{\text{Policy Drift}} \cdot \underbrace{\exp(\Delta\eta(x))}_{\text{Noise}}, \quad (3)$$

where $\Delta\mathcal{L}(x) = \mathcal{L}_\theta(x) - \mathcal{L}_{\theta_{\text{old}}}(x)$ is the noise-free log-ratio proxy, capturing policy drift between the target and behavior policies, and $\Delta\eta(x) = \eta_\theta(x) - \eta_{\theta_{\text{old}}}(x)$ is the net difference in estimation error between the two policy evaluations.

When the log-ratio estimation noise has a non-negligible right tail, exponentiation converts it into a highly skewed ratio distribution with rare but large outliers. For instance, the estimated ratio of a single rollout can explode to magnitudes of $10^5$ (Fig. 8); Appendix D.2 reports a direct noise-

assumption diagnostic on GSM8K. Consequently, for a group of rollouts $\{x_1, \ldots, x_G\}$, the resulting set of importance ratios $\{\hat{\rho}(x_1), \ldots, \hat{\rho}(x_G)\}$ exhibits extremely high variance.

**(ii) Gradient spikes.** We observe that the noise in $\hat{\rho}(x)$ leads to gradient spikes through two mechanisms: individual anomalies and group anomalies.

*Individual Anomalies.* In algorithms like GRPO, clipping is conditional. Specifically, when the advantage is negative ($A < 0$) and the ratio deviates significantly ($\hat{\rho} > 1 + \epsilon$), the objective function simplifies to the unclipped term $\hat{\rho}A$. This design allows the model to take large steps when returning to the trust region. However, in dLLMs, large $\hat{\rho}$ values can be driven by the model-agnostic noise $\Delta\eta(x)$ rather than true policy alignment. Consequently, whenever $A < 0$, there is a probability that a noise-induced outlier results in a massive, unclipped gradient.

*Group Anomalies.* Due to the high variance of the estimator, importance ratios within a group $\{\hat{\rho}_j\}_{j=1}^{G}$ can be simultaneously large or small. Even if individual ratios are capped, the collective fluctuation of the sum $\sum \hat{\rho}_j$ causes the gradient magnitude to oscillate. In Sec. 3.3 and Sec. 4.3, we theoretically and empirically show that these frequent spikes, even bounded, can destabilize the training dynamics.

**(iii) Policy drifts.** When the target policy $\pi_\theta$ undergoes an update driven by a gradient spike, its behavior shifts abruptly, causing the noise-free policy-drift term $\Delta\mathcal{L}(x)$ to increase significantly. As shown in Eq. (3), a larger $\Delta\mathcal{L}(x)$ acts as a multiplier, amplifying the variance of the importance ratios $\{\hat{\rho}_j\}_{j=1}^{G}$ in subsequent steps. This establishes an *instability loop*: estimation noise generates gradient spikes, which induce policy drift; this drift, in turn, exacerbates the variance of future importance ratios. This self-reinforcing loop destabilizes training and leads to reward collapse.

### 3.2 StableDRL

To stabilize training, we propose StableDRL. Our method breaks the instability loop by preventing importance ratio noise from translating into gradient spikes. It consists of two components: unconditional clipping and self-normalization.

**Unconditional clipping.** Throughout the StableDRL update, we use an upper-clipping operator
$$\text{clip}_\epsilon^+(r) := \min\{r, 1 + \epsilon\}, \qquad r > 0.$$
This operator differs from the standard two-sided PPO/GRPO clipping operator used in Eq. (2): StableDRL only suppresses abnormally large estimated ratios, while the positivity of $\hat{\rho}$ provides the lower side. With our default $\epsilon = 5$, this means $0 < \text{clip}_\epsilon^+(\hat{\rho}_j) \leq 6$.

We replace the conditional clipping of GRPO with a strict, unconditional upper bound. We enforce that the positive importance ratio $\hat{\rho}$ is always capped by $\text{clip}_\epsilon^+(\hat{\rho})$, regardless

of the sign of the advantage. Theoretically, this ensures that no individual estimated ratio can exert unbounded leverage.

**Self-normalization.** While unconditional clipping mitigates individual outliers, clipping alone still leaves a random group-scale factor. Specifically, the clipping-only update
$$\frac{1}{G} \sum_{j=1}^{G} \text{clip}_\epsilon^+(\hat{\rho}_j) A_j g_j$$
can oscillate substantially as the clipped group mass $\sum_j \text{clip}_\epsilon^+(\hat{\rho}_j)$ fluctuates, often approaching the upper boundary. This creates a trade-off: a loose upper cap still permits large noise-induced updates, while an overly tight cap obscures useful importance-weight information and slows learning.

To address group-level anomalies, we replace the fixed group-size normalizer $G$ with the realized clipped group mass $\sum_{i=1}^{G} \text{clip}_\epsilon^+(\hat{\rho}_i)$. The StableDRL update is
$$\nabla_\theta \mathcal{J}_{\text{Ours}} = \mathbb{E}\left[\frac{1}{\sum_{i=1}^{G} \text{clip}_\epsilon^+(\hat{\rho}_i)} \sum_{j=1}^{G} \text{clip}_\epsilon^+(\hat{\rho}_j) A_j g_j\right].$$
(4)

Equivalently, Eq. (4) uses normalized coefficients $\alpha_j \propto \text{clip}_\epsilon^+(\hat{\rho}_j)$, replacing the fixed group-size average with a self-normalized clipped importance-weighted update. Sec. 3.3 formalizes the resulting convex-combination view, and Appendix C.5 gives the coordinate-wise bias bound in terms of clipped-normalizer deviation and removed upper-tail mass.

StableDRL is simple yet effective at suppressing gradient spikes, preventing training from entering the instability loop.

### 3.3 Theoretical Analysis

We explain why GRPO becomes unstable when importance ratios are computed from noisy likelihood proxies in dLLMs. Our analysis models a self-reinforcing loop between *estimation noise*, *gradient spikes*, and *policy drift*. We first show that, under GRPO's asymmetric unclipping on negative-advantage samples, the update norm has a nonzero probability of exceeding any threshold $H$. We then show a feedback mechanism: once a spike-induced step increases a drift state, the derived lower bound on the spike probability is nondecreasing for later inner steps on the same rollout group.

The results below should be read under the stated tail-envelope, negative-advantage domination, bounded-direction, and local smoothness conditions. Under these conditions, the lower bound on spike probability is non-decreasing with policy drift, which formalizes a mechanism for self-reinforcing instability rather than an unconditional collapse theorem.

**Notations.** Fix a rollout group $\mathcal{B} = \{x_1, \ldots, x_G\}$ sampled from $\pi_{\theta_{\text{old}}}$, and consider inner updates $\theta_0 = \theta_{\text{old}}, \theta_1, \theta_2, \ldots$ on this *fixed* group. Following Sec. 3.1, let $\hat{\rho}_{i,j} =$

$\exp(\Delta\mathcal{L}_{i,j} + \Delta\eta_{i,j})$ be the estimated importance ratio at inner step $i$, where $\Delta\mathcal{L}_{i,j}$ is the noise-free drift and $\Delta\eta_{i,j}$ is the log-ratio estimation noise. Let $g_{i,j}$ denote the advantage-weighted proxy gradient used in the update. We use a single constant $B$ such that $\|g_{i,j}\| \leq B$ for all inner steps $i$ and samples $j$, which is standard in practice. Define the negative-advantage set $\mathcal{N} = \{j : A_j \leq -a_0\}$ for some $a_0 > 0$, and the drift state $D_i := \max_{j \in \mathcal{N}} \Delta\mathcal{L}_{i,j}$.

**Uniform tail envelope.** To address the non-stationarity of noise across steps, we assume the right tails of $\Delta\eta_{i,j}$ admit a common lower bound (App. C.2). This assumption allows us to lower-bound the spike probability using a time-invariant function of the drift state $D_i$, ensuring that the instability risk is strictly defined by the magnitude of the drift.

**Why GRPO can spike.** Standard GRPO allows *unclipped* multipliers on negative-advantage samples when $A_j < 0$ and $\hat{\rho}_{i,j} > 1 + \epsilon$. We first establish that gradient spikes are statistically inevitable and tightly coupled to the drift state.

**Lemma 3.1** (*Informal*, **existence of drift-dependent spike probability**). *In inner step $i$ of GRPO, for any threshold $H > 0$, there exists a lower bound $P_i(H) \in (0, 1)$ such that*
$$\Pr\left(\|\nabla_\theta \mathcal{J}_{\mathrm{GRPO}}\| \geq H \,\middle|\, D_i\right) \geq P_i(H). \quad (5)$$
*Moreover, under the common tail-envelope condition on $\Delta\eta_{i,j}$, the bound $P_i(H)$ can be chosen as the* same nondecreasing function of $D_i$ for all inner steps *(see App. C.2).*

As the policy drifts ($D_i \uparrow$), the noise margin required to push an importance ratio above any fixed level shrinks. Consequently, large-multiplier outliers become increasingly probable.

**Theorem 3.2** (*Informal*, **self-reinforcing instability loop**). *Consider an inner step $i$ where a gradient spike occurs ($\|\nabla_\theta \mathcal{J}_{\mathrm{GRPO}}\| \geq H$) and the spike is driven by a single negative-advantage outlier that dominates the group update (sufficient conditions are in App. C.2). Then the resulting update increases the drift state:*
$$D_{i+1} \geq D_i. \quad (6)$$
*Consequently, since the bound $P_i(H)$ in Lemma 3.1 is defined via a common tail envelope and is nondecreasing in $D_i$, the next-step spike lower bound cannot decrease on that realized step:*
$$P_{i+1}(H) \geq P_i(H). \quad (7)$$

**Upper clipping alone can saturate.** Unconditional upper clipping prevents unbounded ratio multipliers, but may enter a high-frequency "boundary-hitting" regime near the upper cap.

**Lemma 3.3** (*Informal*, **existence of hitting probability**). *In clipping-only GRPO with upper-clipped weights $w_{i,j} := \mathrm{clip}_\epsilon^+(\hat{\rho}_{i,j})$, the update norm is deterministically bounded by*
$$H_{\max} = (1 + \epsilon)B. \quad (8)$$
*For any threshold $H$ close to $H_{\max}$, there exists a saturation*

probability $Q_i(H)$ that the update hits the upper boundary. Under the same common tail-envelope condition, $Q_i(H)$ can be chosen to be nondecreasing in $D_i$ (see App. C.3).

**Theorem 3.4** (*Informal*, **self-reinforcing hitting loop**). *If at inner step $i$ the update saturates near the upper boundary ($\|\nabla_\theta \mathcal{J}_{\mathrm{UC\text{-}GRPO}}\| \geq H$) and the saturated step induces non-decreasing drift ($D_{i+1} \geq D_i$ under the appendix conditions), then, by monotonicity of $Q_i(H)$ in Lemma 3.3,*
$$Q_{i+1}(H) \geq Q_i(H). \quad (9)$$

Thus, clipping alone can trade rare, unbounded spikes for frequent boundary-saturated updates. This creates a trade-off: a loose upper bound still destabilizes optimization, while an overly tight bound can obscure the importance-weight signal.

**Why StableDRL breaks the loop.** Finally, we show how StableDRL structurally removes the remaining group-scale randomness.

**Theorem 3.5** (**StableDRL**). *Let $w_{i,j} := \mathrm{clip}_\epsilon^+(\hat{\rho}_{i,j})$ be the upper-clipped positive weights. The StableDRL update $\nabla_\theta \mathcal{J}_{\mathrm{Ours}}$ is normalized by the sum of weights. Since $w_{i,j} > 0$, the update always lies in the **convex hull** of the per-sample advantage-weighted update directions $\{g_{i,j}\}$, where $g_{i,j}$ was defined above as the advantage-weighted proxy gradient:*
$$\|\nabla_\theta \mathcal{J}_{\mathrm{Ours}}\| = \left\|\frac{\sum_{j=1}^{G} w_{i,j}\, g_{i,j}}{\sum_{j=1}^{G} w_{i,j}}\right\| \leq \max_j \|g_{i,j}\| \leq B. \quad (10)$$

Unlike clipping alone, self-normalization explicitly divides out the random group-scale factor $\frac{1}{G}\sum_j w_{i,j}$, decoupling the update magnitude from group-level weight fluctuations. This breaks the instability mechanisms formalized in App. C.2 and App. C.3. In Sec. 4.3, we empirically validate these explanations.

### 3.4 Generalization to Block Diffusion

Adapting block diffusion (Wu et al., 2025; Cheng et al., 2025) to RL creates a dilemma between training efficiency and information leakage. Valid likelihood proxy estimation requires conditioning each block strictly on its clean history. Naive iterative implementations are prohibitively slow ($\mathcal{O}(K)$), while standard parallel attention invalidates gradient signals by allowing tokens to "cheat" and attend to their own ground truth.

To resolve this, we introduce *staircase attention*, a structured masking primitive that enables leakage-free, single-pass evaluation. By utilizing a dual-stream input of frozen clean context and corrupted target, the mask enforces strict conditional independence through a unique geometry (Figure 5, Appendix B). A block-lower-triangular "staircase" grants target tokens in block $k$ access to the clean history of preceding blocks $(1 \ldots k-1)$ while mechanically occluding

the current block's ground truth. Simultaneously, a block-diagonal component permits parallel, independent denoising within the target stream. This structure satisfies ELBO requirements within a single computational graph ($\mathcal{O}(1)$), rendering full-parameter RL feasible for long-horizon tasks (formal derivation in Appendix B).

### 3.5 Pratical Implementations

**Score-function surrogates.** Since dLLMs lack a tractable $\nabla_\theta \log \pi_\theta$, StableDRL reweights gradient directions provided by stable score surrogates, current state-of-the-art dLLM RL methods SPG (Yang et al., 2025b; Ou et al., 2025a; Wang et al., 2025a). For full-attention dLLMs, we implement *block-wise masking* (Wang et al., 2025a) by sampling structured mask blocks consistent with the inference denoising schedule. For block diffusion, we sample random positions to mask since the staircase attention runs in $\mathcal{O}(1)$. The bounded-direction assumption in Sec. 3.3 is a theoretical regularity condition on advantage-weighted per-sample directions. We do not add explicit per-sample gradient clipping in the implementation because that would require separate forward-backward passes for each rollout sample; empirically, the observed failures are driven by noisy ratio outliers and group-scale fluctuations.

**Numerically stable log-space weights.** Direct computation of Eq. (4) is numerically unstable due to the exponentiation of noisy log-ratios. We strictly compute weights in log-space. Let
$$\ell_j = \widehat{\mathcal{L}}_\theta(x_j) - \widehat{\mathcal{L}}_{\theta_{\text{old}}}(x_j), \qquad \tilde{\ell}_j = \min\{\ell_j, \log(1+\epsilon)\}.$$
We compute the normalized coefficients as
$$\alpha_j = \exp\big(\tilde{\ell}_j - \text{LSE}(\{\tilde{\ell}_k\}_{k=1}^G)\big),$$
equivalently $\alpha_j = \text{softmax}(\tilde{\ell})_j$, where $\text{LSE}(\cdot)$ is the Log-Sum-Exp function. This *clip-then-softmax* approach preserves numerical precision even when raw probability ratios would underflow or overflow, ensuring stable optimization in mixed-precision training.

## 4 Experiments

We evaluate StableDRL on two diffusion architectures: full-attention masked diffusion (`LLaDA-8B-Instruct`) and semi-autoregressive block diffusion (`SDAR-8B`). Our experiments address three questions: (i) whether StableDRL improves reasoning performance, (ii) whether it fixes the diagnosed spike-drift mechanism, and (iii) whether the gains reflect stabilization rather than slower optimization or favorable hyperparameters. We first report the main reasoning results, then validate the stability mechanism, and finally study ablations and clipping sensitivity.

**Experimental setup.** For both `LLaDA-8B-Instruct` and `SDAR-8B`, We perform RL fine-tuning using the AdamW optimizer with a learning rate of $1.0 \times 10^{-6}$. To ensure optimization stability while maintaining sample effi-

ciency, we set the unconditional importance weight clipping threshold to $\epsilon = 5$ by default. For a comprehensive description of the training infrastructure, model configurations, and hyperparameters, please refer to Appendix D.1.

### 4.1 Full-Attention Diffusion Results

**Experimental setup.** We follow the experimental protocol of ESPO (Ou et al., 2025a) and SPG (Wang et al., 2025a), which build on the D1 and WD1 setup (Zhao et al., 2025b; Tang et al., 2025): benchmarks include GSM8K (Cobbe et al., 2021), MATH500 (Lightman et al., 2023), Countdown (Pan et al., 2025), and Sudoku (Arel, 2025), with the same train/test splits and evaluation procedure. Concretely, we evaluate at generation lengths $\{128, 256, 512\}$ and use confidence-based semi-autoregressive decoding with block size 32 for both RL rollouts and evaluation. We set $\epsilon = 5$ for the best performance.

**Baselines.** We compare StableDRL against a representative suite of reinforcement learning algorithms for dLLMs. Baselines include **D1** (Zhao et al., 2025b) and **UniGRPO** (Yang et al., 2025b), which adapt the GRPO framework by approximating the intractable log-likelihood via one-step unmasking or MC estimation of the ELBO. We also include **WD1** (Tang et al., 2025), which formulates a weighted policy optimization objective to avoid direct likelihood estimation. Finally, we benchmark against **SPG** (Wang et al., 2025a), which mitigates gradient bias by sandwiching the policy objective between a tractable Evidence Upper Bound (EUBO) for negative rewards and the ELBO for positive rewards. All methods are initialized from the `LLaDA-8B-Instruct` (Nie et al., 2025a).

**Enabling stable full fine-tuning.** Unlike ESPO or SPG, which mitigate instability via LoRA or early stopping, we *fully fine-tune* `LLaDA-8B-Instruct` by explicitly suppressing the gradient spikes that typically destabilize training. This allows StableDRL to optimize the entire model backbone, better unlocking the latent reasoning capabilities of the dLLM. Notably, while our RL training is conducted at a sequence length of 256 tokens, the resulting model achieves consistent, high performance across all evaluated generation lengths (128 to 512 tokens). This suggests that stable, full-parameter reinforcement learning fosters superior length generalization compared to parameter-efficient or variance-constrained alternatives.

**State-of-the-art performance.** Table 1 demonstrates that StableDRL establishes a new state-of-the-art by achieving the highest average accuracy across all decoding budgets. Specifically, in complex reasoning (MATH500), it secures an average accuracy of 41.8%, outperforming ESPO and SPG, with a notable +5.2% margin over SPG at the 256-token budget. In long-horizon planning (Countdown), StableDRL overcomes the off-policy drift that plagues SPG, de-

*Table 1.* **State-of-the-Art Reasoning Performance on LLaDA-8B-Instruct.** We report pass@1 accuracy under three decoding budgets ($N \in \{128, 256, 512\}$) and the mean performance (**Avg**) for each dataset. **Bold** denote the best result and underline the second best. StableDRL achieves the highest average accuracy on all four benchmarks, demonstrating superior consistency across generation lengths.

| Model / Seq Len | GSM8K | | | | MATH500 | | | | Countdown | | | | Sudoku | | | |
|---|---|---|---|---|---|---|---|---|---|---|---|---|---|---|---|---|
| | 128 | 256 | 512 | **Avg** | 128 | 256 | 512 | **Avg** | 128 | 256 | 512 | **Avg** | 128 | 256 | 512 | **Avg** |
| LLaDA-8B-Inst. | 69.5 | 77.2 | 79.8 | 75.5 | 28.2 | 32.4 | 34.6 | 31.7 | 18.8 | 16.8 | 16.8 | 17.5 | 5.7 | 27.7 | 26.2 | 19.9 |
| LLaDA-1.5 | 70.4 | 80.5 | 81.9 | 77.6 | 26.8 | 32.2 | 35.8 | 31.6 | 31.9 | 21.1 | 21.5 | 24.8 | 7.4 | 26.9 | 29.0 | 21.1 |
| D1 | 72.2 | 80.6 | 81.3 | 78.0 | 31.4 | 36.0 | 39.4 | 35.6 | 30.9 | 30.9 | 34.4 | 32.1 | 7.2 | 32.5 | 29.3 | 23.0 |
| WD1 | 74.6 | 81.5 | 83.0 | 79.7 | 31.0 | 37.4 | 39.0 | 35.8 | 48.8 | 52.3 | 50.8 | 50.6 | 33.1 | 32.1 | 22.5 | 29.2 |
| UniGRPO | 74.9 | 82.5 | 82.7 | 80.0 | 32.4 | 37.4 | 39.4 | 36.4 | 44.5 | 43.0 | 57.0 | 48.2 | 59.0 | 67.0 | 62.9 | 63.0 |
| ESPO | 80.0 | 82.3 | 83.7 | 82.0 | 36.0 | 39.0 | 43.4 | 39.5 | **81.6** | 82.0 | 79.3 | 81.0 | **92.7** | 84.7 | 80.5 | 86.0 |
| SPG w/ Mixture | 78.5 | 86.1 | 84.5 | 83.0 | 33.4 | 40.0 | 41.8 | 38.4 | 68.8 | 70.7 | 70.3 | 69.9 | 82.9 | **94.0** | **93.1** | 90.0 |
| **StableDRL (Ours)** | **80.2** | **86.2** | **86.3** | **84.2** | **36.2** | **45.2** | **44.0** | **41.8** | 81.3 | **84.4** | **84.8** | **83.5** | 91.9 | 92.4 | 90.1 | **91.5** |

livering a massive +13.7% gain at 256 tokens to reach 84.4%. Furthermore, unlike baselines such as ESPO that fluctuate significantly on consistency tasks, StableDRL maintains robustness across all lengths, achieving top average scores on both GSM8K (84.2%) and Sudoku (91.5%). These results confirm that resolving "Noise-Drift" instability is critical for scaling RL in dLLMs.

**Matched fairness and optimization speed.** Section 4.3 reports matched GSM8K reruns that separate StableDRL's stabilization effect from slower early optimization and LoRA containment.

## 4.2 Generalization to Block Diffusion

To demonstrate the architectural generality of our framework, we instantiate StableDRL on the *SDAR-8B-Chat* block diffusion model (Cheng et al., 2025). We utilize Staircase Attention (Sec. 3.4) to enable scalable, leakage-free proxy estimation during training.

**Experimental setup.** We follow TraceRL's training and evaluation conventions for the *SDAR-8B-Chat* (Cheng et al., 2025) model ($B = 4$). Each RL sampling iteration generates 16 trajectories per prompt using *dynamic sampling* ($T = 0.9$, temp 1.0). We train on the selected MATH training data (Lightman et al., 2023). Evaluation uses both (i) *static* (greedy block-wise) and (ii) *dynamic* (temperature 1.0) sampling.

**Baselines.** We benchmark against the supervised base model **SDAR-8B**, **Trado** (Wang et al., 2025b) using the model with only TraceRL training for fairness, and the autoregressive **Qwen3** (Yang et al., 2025a) (4B and 8B Base) to contextualize performance against standard LLMs. We exclude **DiRL** (Zhu et al., 2026) from our comparison, as it utilizes a fundamentally different data regime and a complex two-stage training pipeline.

*Table 2.* **Block Diffusion Reasoning Performance.** Pass@1 accuracy on MATH500, GSM8K, and AIME, comparing StableDRL (SDAR-8B backbone) against AR baselines (Qwen3) and prior Block Diffusion methods. StableDRL notably outperforms the strong Qwen3-8B AR model on the rigorous AIME benchmark.

| Method | MATH500 | GSM8K | AIME 24 |
|---|---|---|---|
| *Autoregressive (AR) Baselines* | | | |
| Qwen3-4B | 74.1 | 90.7 | 12.9 |
| Qwen3-8B | 78.4 | **92.8** | 10.0 |
| *Block Diffusion (Dynamic Sampling)* | | | |
| SDAR-8B (Base) | 70.6 | 90.4 | 8.3 |
| Trado | 75.0 | 91.2 | 11.0 |
| **StableDRL (Ours)** | 77.8 | 92.1 | 13.3 |
| *Block Diffusion (Static Sampling)* | | | |
| SDAR-8B (Base) | 75.4 | 91.1 | 11.8 |
| Trado | 78.5 | 92.3 | 13.3 |
| **StableDRL (Ours)** | **79.2** | 92.4 | **16.7** |

**Performance analysis.** Table 2 reports the comparative results. StableDRL consistently outperforms prior block diffusion methods. Notably, on the rigorous *AIME 2024* benchmark, StableDRL achieves *16.7%* (Static), significantly surpassing the base model (11.8%), Trado (13.3%), and even the *autoregressive Qwen3-8B* (10.0%). This indicates that stable on-policy RL can unlock reasoning capabilities often dormant in supervised baselines. Furthermore, while Trado degrades significantly under dynamic sampling (dropping to 11.0% on AIME), StableDRL maintains superior robustness (*13.3%*), indicating it effectively shapes the full probability landscape rather than merely optimizing the mode.[1] StableDRL operates on importance-ratio estimation and is not tied to a specific reward form. In this paper, we focus on verifiable language reasoning and puzzle rewards; broader

---

[1] To enable computationally feasible training, we employ custom `JetEngine` (Cheng et al., 2025) inference kernels.

rewards and molecular or polymer diffusion RL require domain-specific evaluation and are left for future work.

### 4.3 Empirical Verification of Instability and Stabilization Mechanisms

We verify the mechanism in Sec. 3.3 from three complementary perspectives. First, we test whether the small-budget MC ELBO proxies used in practice indeed induce heavy-tailed ratio noise. Second, we check whether reward collapse is accompanied by gradient-spike amplification in a same-task diagnostic. Third, we compare full fine-tuning and LoRA under matched settings to separate StableDRL's stabilization effect from slower optimization or parameter-efficient containment.

**Validation of the noise assumption.** The instability analysis in Sec. 3.3 assumes that log-ratio estimation noise has a non-negligible right tail. To test whether this assumption is relevant under practical dLLM likelihood estimation, we estimate a 1000-sample MC ELBO as an oracle on GSM8K rollouts and bootstrap small-budget MC proxies with $m \in \{2, 5, 10\}$. Table 3 shows that the resulting ratio noise is substantially heavier-tailed than a Gaussian baseline in the small-$m$ regime used by practical RL training. At $m = 2$, the noise has excess kurtosis 6.26 and $\Pr(|\eta| > 3\sigma) = 0.031$, an order of magnitude above the Gaussian reference of 0.003. The induced importance ratios also exhibit extreme upper-tail behavior, with $\rho_{\max} = 47.1$ and $\rho_{99} = 7.4$. Increasing $m$ reduces but does not eliminate the tail effect. These results support the practical relevance of the tail-envelope assumption and explain why direct GRPO-style ratio weighting can be fragile when ratios are estimated from small-budget MC ELBO proxies.

*Table 3.* **Noise-assumption validation on GSM8K.** Small-budget MC ELBO proxies are heavy-tailed and induce extreme importance ratios.

| Metric | $m=2$ | $m=5$ | $m=10$ | **Gaussian** |
|---|---|---|---|---|
| Excess kurtosis | 6.26 | 2.70 | 1.34 | 0 |
| $\Pr(|\eta| > 2\sigma)$ | 0.072 | 0.048 | 0.048 | 0.046 |
| $\Pr(|\eta| > 3\sigma)$ | 0.031 | 0.014 | 0.010 | 0.003 |
| $\rho_{\max}$ | 47.1 | 11.0 | 4.5 | – |
| $\rho_{99}$ | 7.4 | 3.2 | 2.2 | – |
| Dominance | 0.80 | 0.57 | 0.45 | $1/m$ |
| Dominance($|\eta| > p95$) | 0.90 | 0.62 | 0.42 | $1/m$ |

**Training manifestation of the instability loop.** We next verify that the predicted instability appears during actual full-parameter training. Since Appendix Fig. 7 is a qualitative visualization, we use the matched Countdown statistics in Table 4 as the primary main-text evidence. After collapse, the mean training reward decreases from 1.59 to 1.08, while the peak gradient norm increases from 10.2 to 24.5 and the mean spike rate rises from 0.15 to 0.30. This supports the

spike-drift-collapse mechanism: collapse is accompanied not by a benign reward fluctuation, but by amplified gradient spikes and a higher frequency of unstable updates.

*Table 4.* **Same-task Countdown collapse consistency.** Gradient spikes and spike frequency increase after collapse under matched Countdown diagnostics.

| Metric | Pre-collapse | Post-collapse |
|---|---|---|
| Mean train reward | 1.59 | 1.08 |
| Peak gradient norm | 10.2 | 24.5 |
| Mean spike rate | 0.15 | 0.30 |

**Full fine-tuning versus LoRA containment.** Finally, we test whether StableDRL's gains come from slower optimization or from avoiding full-parameter instability. Table 5 reports matched GSM8K reruns under the same backbone, rollout budget, decoding setup, optimizer, learning rate, and training horizon. At 200 steps, full-parameter StableDRL is comparable to ESPO and SPG, achieving 82.2 versus 82.6 and 82.1 accuracy, respectively. Thus, StableDRL does not gain by simply learning more slowly. The difference emerges during continued training: under full fine-tuning, ESPO and SPG collapse before 1000 steps, while StableDRL remains stable and improves to 86.2. LoRA makes all methods more stable by restricting the trainable subspace, but it acts as a containment mechanism rather than solving the full-parameter instability. Full-parameter StableDRL achieves the best overall result, slightly exceeding the strongest LoRA baseline, while preserving the ability to optimize the entire backbone.

*Table 5.* **Matched GSM8K reruns under full fine-tuning and LoRA.** StableDRL is comparable early and remains stable under continued full fine-tuning.

| Regime | Method | Acc.@200 | Acc.@1000 | Grad norm |
|---|---|---|---|---|
| Full FT | StableDRL | 82.2 | 86.2 | 1.90 |
| Full FT | ESPO | 82.6 | N/A | 5.00 |
| Full FT | SPG | 82.1 | N/A | 3.96 |
| LoRA | StableDRL | 78.0 | 84.6 | 0.32 |
| LoRA | ESPO | 79.1 | 82.3 | 0.58 |
| LoRA | SPG | 78.3 | 86.1 | 0.49 |

Appendix D.6 provides the full diagnostic curves, and Appendix D.7 visualizes the joint distribution of importance weights and gradient norms.

### 4.4 Ablation Studies

**Dissecting the stability mechanisms.** To verify the contributions of Unconditional Clipping and Group Self-Normalization, we analyze the training dynamics on Countdown (Figure 3). We observe that removing unconditional clipping leads to rapid training failure, as noise-induced

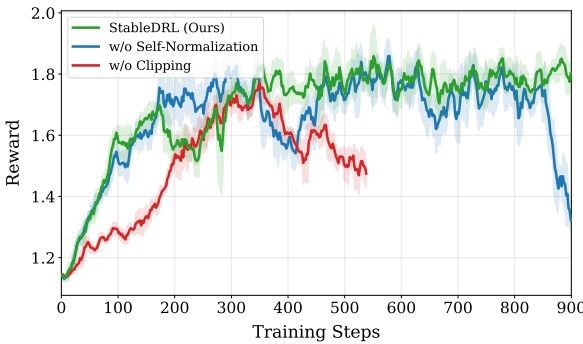

*Figure 3.* **Deconstructing Stability Mechanisms.** On Countdown, clipping without self-normalization remains high-variance, and self-normalization without clipping collapses when noise outliers dominate. StableDRL combines both controls and yields monotonic stability.

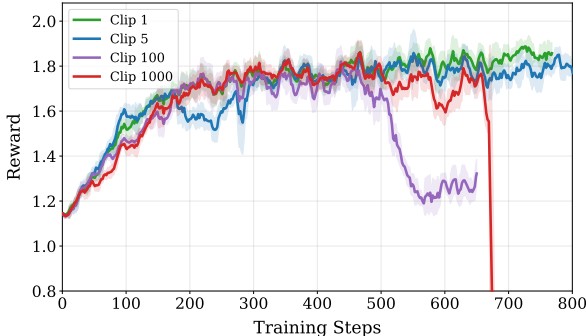

*Figure 4.* **Sensitivity to Trust Region Size** ($\epsilon$). Small thresholds remain stable, with $\epsilon = 5$ giving the best bias-variance trade-off; very loose thresholds allow noisy-ratio outliers to re-emerge and destabilize training.

outliers dominate the convex combination of advantage-weighted update directions. Conversely, removing self-normalization while retaining clipping causes the aggregated update magnitude to oscillate substantially within the bounded range, often approaching the upper boundary due to estimation noise, distorting the AdamW momentum history and eventually leading to reward collapse. Only StableDRL, which combines magnitude bounding with geometric constraints, yields a stable and monotonic learning curve.

**Sensitivity to trust region tightness** ($\epsilon$). We further analyze the trade-off between stability and learning speed across varying clipping thresholds (Figure 4). Our default $\epsilon = 5$ clips ratios to $0 < \rho < 6$ after upper clipping while preserving more learning signal than $\epsilon = 1$; $\epsilon = 1$ is stable but slower. However, very loose thresholds such as 100 or 1000 allow noisy-ratio outliers to reappear and can destabilize training before clipping meaningfully protects the update.

## 5 Related Work

**Positioning.** StableDRL connects RL post-training, discrete diffusion LMs, and robust importance weighting. Policy-gradient RL and RLHF underpin modern LM alignment (Williams, 1992; Schulman et al., 2017; Ouyang et al., 2022), and verifiable-reward RL has driven strong reasoning gains in AR LMs (Shao et al., 2024; DeepSeek-AI, 2025). For dLLMs, prior work improves diffusion modeling or likelihood proxies (Austin et al., 2021; Sahoo et al., 2024; Nie et al., 2025b; Cheng et al., 2025), and recent RL methods such as D1 and ESPO adapt GRPO-style training to intractable dLLM probabilities (Zhao et al., 2025b; Ou et al., 2025a). StableDRL instead focuses on the resulting optimization failure mode: noisy proxy ratios create spike-drift collapse during full-parameter RL.

**Stabilization perspective.** Our clipped self-normalized update is related to truncated or self-normalized importance sampling and off-policy correction methods (Ionides, 2008; Owen, 2013; Espeholt et al., 2018; Munos et al., 2016). The distinction is that, in dLLM RL, even on-policy ratios are noisy estimates from small-budget diffusion likelihood proxies. Recent LM stabilizers address ratio control, staleness, or train-test mismatch (Li et al., 2025; Chen et al., 2026); StableDRL targets the dLLM-specific combination of exponentiated proxy noise and group-scale fluctuation. Appendix A gives a fuller comparison.

## 6 Conclusion

This paper studies the instability of Group Relative Policy Optimization (GRPO) when applied to discrete diffusion large language models. We identify that GRPO instability in dLLMs stems from the noisy Monte Carlo importance ratio estimation, which triggers a self-reinforcing instability loop of gradient spikes and policy drift. To break this loop, we propose StableDRL, which employs unconditional clipping and self-normalization to eliminate spikes. Extensive experiments demonstrate that our proposed approach effectively stabilizes the training and significantly unlocks the reasoning potential of dLLMs. Beyond the empirical gains, our analysis gives a practical criterion for dLLM RL: robust updates must jointly control ratio outliers and prompt-group scale without changing the GRPO interface.

**Limitations.** Our experiments focus mainly on language reasoning and puzzle benchmarks. Although StableDRL is conceptually applicable to discrete diffusion domains with noisy likelihood proxies, molecular and polymer RL require domain-specific rewards, constraints, and evaluation protocols beyond the scope of this work. Full-parameter dLLM RL is compute-intensive, and longer-horizon tasks, broader reward models, and larger-scale multi-seed studies remain important future directions.

## Impact Statement

This paper presents StableDRL, a framework that stabilizes full-parameter reinforcement learning for discrete diffusion large language models (dLLMs). By addressing the "Noise-Drift" instability inherent in estimating importance ratios, our work facilitates the development of more robust and reliable non-autoregressive AI systems.

A primary contribution of this work is the mitigation of training collapse. By ensuring stable optimization, StableDRL significantly reduces the computational waste associated with failed experiments and divergent training runs. Furthermore, by unlocking the reasoning capabilities of dLLMs, this work supports the adoption of non-autoregressive architectures, which offer the potential for more energy-efficient, parallelized inference compared to standard autoregressive models.

We demonstrate that stable RL effectively unlocks the latent reasoning potential of dLLMs in domains such as mathematics and logic. While these capabilities have positive applications in scientific discovery, education, and automated problem-solving, they also carry the inherent dual-use risks associated with any advanced generative AI, such as the potential for generating misleading content or automating malicious tasks.

Crucially, the stability provided by StableDRL is a prerequisite for effective safety alignment. By enabling reliable on-policy optimization, our method provides the necessary tooling to apply safety constraints and human feedback (e.g., RLHF) to diffusion language models, potentially making them more controllable and safer than their unstable predecessors.

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

# A   Extended Related Work

**RL post-training for LMs.** Policy-gradient RL underpins modern alignment and post-training pipelines (Williams, 1992; Schulman et al., 2015; 2017). RLHF popularized preference-based alignment for AR LMs (Ouyang et al., 2022), while RL with verifiable rewards has shown strong gains for mathematical reasoning and long-form solutions (Shao et al., 2024; DeepSeek-AI, 2025). In the AR context, importance-ratio correction (Team et al., 2025; Zheng et al., 2025) mainly targets policy staleness between behavior and target policies.

**Discrete diffusion language models.** StableDRL builds on masked/discrete diffusion language models and sequence-level likelihood proxies rather than changing the diffusion generator itself. Representative discrete diffusion formulations and dLLM scaling work include D3PMs (Austin et al., 2021), continuous-time discrete denoising (Campbell et al., 2022), score-entropy discrete diffusion (Lou et al., 2024), simplified masked diffusion (Shi et al., 2024), MDLM (Sahoo et al., 2024), LLaDA (Nie et al., 2025b), and SDAR (Cheng et al., 2025).

**RL for diffusion LMs.** Other methods such as LLaDA 1.5 (Zhu et al., 2025) study off-policy RL for diffusion LMs, while on-policy approaches such as D1 (Zhao et al., 2025b) and ESPO (Ou et al., 2025a) more closely match the rollout setting studied here. Meanwhile, MDPO (He et al., 2025) models the diffusion process as a formal Markov decision process, which can be computationally expensive and challenging to scale to large-parameter models.

**Robust importance weighting.** Our clipped self-normalized update is related to truncated and self-normalized importance sampling (Ionides, 2008; Hesterberg, 1988; Owen, 2013; Elvira & Martino, 2021), V-trace/IMPALA (Espeholt et al., 2018), and Retrace (Munos et al., 2016). The setting differs because in dLLM RL the ratio itself is a noisy likelihood proxy computed from small-budget diffusion estimates, even in an on-policy update.

**Language-model RL stabilization.** Recent LM stabilization methods such as ST-PPO (Li et al., 2025) and concurrent LCO (Chen et al., 2026) address instability from ratio control, policy staleness, or training-inference mismatch. Stable-DRL instead targets dLLM-specific spike-drift collapse caused by exponentiated proxy-estimation noise and group-scale normalization.

# B   Details on Staircase Attention and Proxy Estimation

In this section, we provide the theoretical details for adapting Reinforcement Learning to Block Diffusion models. We discuss the Monte Carlo estimation of the objective, the efficiency-leakage dilemma, and the formal construction of the Staircase Attention mask.

## B.1   Monte Carlo Estimation of ELBO

For a fixed context $c$ and sequence $x$, we estimate the lower-bound likelihood proxy by sampling $m$ perturbations. Let $\xi = (t, M_t)$ collect the internal diffusion randomness (time and mask). A generic Monte Carlo (MC) estimator of the ELBO takes the form:

$$\widehat{\mathcal{L}}_{\text{ELBO}}(x \mid c; \theta) = \frac{1}{m} \sum_{\tau=1}^{m} \Big[ w(t_\tau) \sum_{i=1}^{n} \mathbf{1}(M_{t_\tau}^i = 1) \log \pi_\theta(x^i \mid x_{t_\tau}^{(\tau)}, c) \Big], \tag{11}$$

where $x_{t_\tau}^{(\tau)}$ is produced by the forward process using $(t_\tau, M_{t_\tau})$. In standard full-attention models, the conditional $\log \pi_\theta$ is computed under a full bidirectional mask. However, for block diffusion, we must enforce block-wise conditional independence to ensure the estimator remains a valid lower bound.

## B.2   The Efficiency-Leakage Dilemma

For a sequence divided into $K$ blocks, an exact ELBO estimate requires conditioning each block $B_k$ strictly on its clean history $x_{<B_k}$.

- **Naive Iterative Implementation** ($O(K)$)**:** This necessitates $K$ separate forward passes, masking future tokens sequentially. For long sequences (e.g., 64 blocks), this increases the training cost linearly, rendering iterative RL prohibitively expensive.

- **Standard Single-Pass (Leakage):** Conversely, standard bidirectional attention allows all tokens to attend to the full sequence. If applied naively in a single pass, denoising tokens in block $B_k$ would attend to the ground-truth representations of their own block, mathematically invalidating the variational bound and the gradient signal.

In our block-diffusion evaluation setting with 8 groups, 16 rollouts per group, and MC budget 2, DDP staircase evaluation uses about 15GB of memory and roughly 20 seconds per evaluation batch. An iterative leakage-free implementation has a similar memory footprint, but takes about 2 minutes because blocks must be evaluated sequentially.

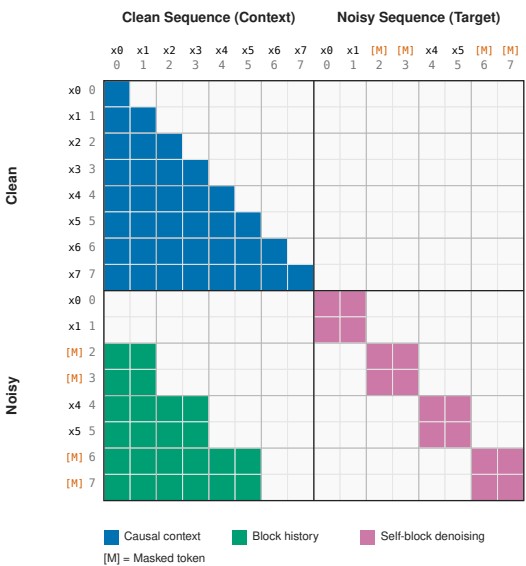

*Figure 5.* **Staircase Attention for Efficient Proxy Estimation.** To evaluate the ELBO for block diffusion in a single pass ($O(1)$), we use a dual-stream construction. The Clean Context (top rows) provides immutable history. The Corrupted Target stream (bottom rows) uses a "staircase" mask ($M_{\text{STAIR}}$, bottom-left) to attend to valid history without peeking at the ground truth of the current block. The target self-attention ($M_{\text{INTRA}}$, bottom-right) is block-diagonal, ensuring independent parallel denoising.

## B.3  Dual-Stream Input and Mask Construction

To achieve $O(1)$ evaluation without leakage, we employ a dual-stream ("2L") input construction. We concatenate a *clean context stream* $x_{\text{ctx}}$ (frozen history) and a *corrupted target stream* $x_{\text{tgt}}$ (containing mask tokens). Let $\tilde{x} = [x_{\text{ctx}}; x_{\text{tgt}}]$ be the combined input of length $2n$.

We define a composite attention mask $M \in \{0, 1\}^{2n \times 2n}$ that enforces strict block-causal dependency:

$$M = \begin{bmatrix} M_{\text{CAUSAL}} & \mathbf{0} \\ M_{\text{STAIR}} & M_{\text{INTRA}} \end{bmatrix}. \tag{12}$$

The components are defined as follows:

1. **Top-Left ($M_{\textbf{CAUSAL}}$):** Standard causal mask for the clean context stream (Blue regions in Figure 5).

2. **Top-Right (0):** Zero matrix. The clean context cannot attend to the noisy target.

3. **Bottom-Right ($M_{\textbf{INTRA}}$):** A block-diagonal mask where $(M_{\text{INTRA}})_{ij} = 1$ iff target positions $i$ and $j$ belong to the same block. This corresponds to the **Pink** regions in Figure 5 and enables intra-block denoising.

4. **Bottom-Left ($M_{\textbf{STAIR}}$):** The strictly block-lower-triangular component, corresponding to the **Green** regions in Figure 5. For a target token in block $k$ and a context token in block $l$:

$$(M_{\text{STAIR}})_{k,l} = \begin{cases} 1 & \text{if } l < k \quad \text{(Context: Attend to history)} \\ 0 & \text{if } l \geq k \quad \text{(Context: Occlude current/future)} \end{cases} \tag{13}$$

This construction allows us to compute gradients for all $K$ blocks simultaneously while mathematically preserving the autoregressive factorization required by the objective.

# C   Proof of Main Results

## C.1   Formal theorem statements for Sec. 3.3

This subsection presents the formal statements of the instability mechanisms identified in Section 3.3. Theorems C.1–C.3 outline our theoretical framework in three logical steps. First, **Theorem C.1** formally characterizes the drift–spike feedback loop inherent to standard GRPO. Second, **Theorem C.2** states that unconditional upper clipping, while mitigating unbounded spikes, may lead to frequent upper-boundary saturation. Finally, **Theorem C.3** establishes that self-normalization structurally resolves the remaining random group-scale factor. Detailed proofs are provided in subsequent subsections.

**Mathematical setup.** Fix a behavior policy $\theta_{\text{old}}$ and a rollout group $\mathcal{B} = \{x_1, \ldots, x_G\}$ sampled from $\pi_{\theta_{\text{old}}}$. GRPO performs updates $\theta_0 = \theta_{\text{old}}, \theta_1, \theta_2, \ldots$ on this same fixed group. Write the estimated importance ratio on sample $x_j$ at step $i$ as
$$\hat{\rho}_{i,j} = \exp(\Delta\mathcal{L}_{i,j} + \Delta\eta_{i,j}),$$
where $\Delta\mathcal{L}_{i,j} = \mathcal{L}_{\theta_i}(x_j) - \mathcal{L}_{\theta_{\text{old}}}(x_j)$ is the noise-free drift and $\Delta\eta_{i,j}$ is the corresponding log-ratio noise term. Let $\widehat{A}_j$ be the fixed, group-relative advantage, and define the negative set
$$\mathcal{N} = \{j : \widehat{A}_j \leq -a_0\} \quad \text{for some } a_0 > 0.$$
Define the drift state, within-negative spread, and drift-maximizer index
$$D_i = \max_{j \in \mathcal{N}} \Delta\mathcal{L}_{i,j}, \qquad S_i = D_i - \min_{j \in \mathcal{N}} \Delta\mathcal{L}_{i,j}, \qquad j^\dagger \in \arg\max_{j \in \mathcal{N}} \Delta\mathcal{L}_{i,j}. \tag{14}$$
Finally, let $\widehat{g}_{\text{GRPO},i}$ denote the implemented GRPO update direction at step $i$.

**Theorem C.1** (GRPO drift–spike feedback loop). *Assume the standing Conditions (C1)–(C5) in Appendix C.2. Fix any spike threshold $H > 0$ and define*
$$u_H := \max\left\{1 + \epsilon, \ \frac{GH}{(1-\lambda)a_0 b_0}, \ u_0\right\}, \tag{15}$$
*where $u_0$ is the deterministic constant defined in Lemma C.6. Define the spike-probability lower bound*
$$P_i(H) := \frac{1}{2} \bar{F}(\log u_H - D_i). \tag{16}$$
*Then, almost surely (conditioning on $\mathcal{F}_{i-1}$, i.e., the current inner iterate and the fixed rollout group),*
$$\mathbb{P}(\|\widehat{g}_{\text{GRPO},i}\| \geq H \mid \mathcal{F}_{i-1}) \geq P_i(H), \qquad a.s. \tag{17}$$
*and $P_i(H)$ is a nondecreasing function of the drift state $D_i$.*

*Moreover, on any realized step where a single negative-advantage outlier dominates the group update and the local smoothness/geometry conditions in Appendix C.2 hold for that realized update, there exist step-dependent scalars $c_{\text{sup},i} > 0$ and $c_{\text{amp},i} \in \mathbb{R}$ and indices $j^\star \in \mathcal{N}$ and $j^\diamond \in \mathcal{N} \setminus \{j^\star\}$ such that*
$$\mathcal{L}_{\theta_i}(x_{j^\star}) - \mathcal{L}_{\theta_{i+1}}(x_{j^\star}) \geq c_{\text{sup},i} \frac{\hat{\rho}_{i,j^\star}}{G}, \tag{18}$$
$$D_{i+1} \geq D_i + (c_{\text{amp},i}\hat{\rho}_{i,j^\star} - S_i).$$
*In particular, if $c_{\text{amp},i}\hat{\rho}_{i,j^\star} \geq S_i$, then $D_{i+1} \geq D_i$, hence*
$$P_{i+1}(H) \geq P_i(H) \qquad \text{(on that realized step).} \tag{19}$$

**Theorem C.2** (Boundary saturation under upper clipping). *Let $g_{i,j} := \widehat{A}_j \nabla_\theta \mathcal{L}_{\theta_i}(x_j)$ and assume $\|g_{i,j}\| \leq B$ (Condition (C1)). Define the upper-clipped positive weight $w_{i,j} := \min(\hat{\rho}_{i,j}, 1 + \epsilon)$ and the clipping-only direction $\widehat{g}_{\text{clip},i} := \frac{1}{G} \sum_{j=1}^{G} w_{i,j} g_{i,j}$.*

*First, clipping prevents unbounded spikes: deterministically, $\|\widehat{g}_{\text{clip},i}\| \leq (1 + \epsilon)B$.*

*Second, drift still increases the frequency of hitting the upper clipping boundary. Let $j^\dagger \in \arg\max_{j \in \mathcal{N}} \Delta\mathcal{L}_{i,j}$ be a drift-maximizer in the negative set. Then, almost surely,*
$$\mathbb{P}(\hat{\rho}_{i,j^\dagger} \geq 1 + \epsilon \mid \mathcal{F}_{i-1}) \geq \bar{F}(\log(1 + \epsilon) - D_i), \tag{20}$$
*and the right-hand side is nondecreasing in $D_i$.*

*Under the additional dominance event in Lemma C.14 and the same local smoothness/geometry conditions used in Appendix C.2, there exists a step-dependent scalar $c_{\text{amp},i} \in \mathbb{R}$ such that*
$$D_{i+1} \geq D_i + (c_{\text{amp},i}(1 + \epsilon) - S_i).$$
*Thus clipping alone can replace rare extreme spikes with frequent boundary-saturated updates once drift becomes large.*

**Theorem C.3** (Self-normalization removes the random group-scale factor). *With upper-clipped positive weights $w_{i,j} :=$*

$\min(\hat{\rho}_{i,j}, 1+\epsilon)$, *define the self-normalized direction*

$$\widehat{g}_{\mathrm{sn},i} := \frac{\sum_{j=1}^{G} w_{i,j} g_{i,j}}{\sum_{j=1}^{G} w_{i,j}},$$

$$\widehat{g}_{\mathrm{clip},i} = \left(\frac{1}{G} \sum_{j=1}^{G} w_{i,j}\right) \widehat{g}_{\mathrm{sn},i}. \tag{21}$$

*Since $w_{i,j} > 0$ (because $\hat{\rho}_{i,j} = \exp(\cdot) > 0$) and $\sum_j w_{i,j} > 0$, the coefficients $w_{i,j}/\sum_k w_{i,k}$ form a convex combination, so $\widehat{g}_{\mathrm{sn},i}$ always lies in the convex hull of $\{g_{i,j}\}_{j=1}^{G}$. In particular, if $\|g_{i,j}\| \leq B$ then deterministically $\|\widehat{g}_{\mathrm{sn},i}\| \leq B$. Thus self-normalization explicitly divides out the random group-scale factor $\frac{1}{G}\sum_j w_{i,j}$ that remains under clipping-only.*

## C.2   Proof of Theorem C.1

We present the proof of Theorem C.1. For clarity, we first state the necessary setup and assumptions.

**Deterministic proxy gradients and GRPO effective weights.** Define deterministic proxy gradients
$$h_{i,j} := \nabla_\theta \mathcal{L}_{\theta_i}(x_j), \qquad g_{i,j} := \widehat{A}_j h_{i,j}.$$
We write the implemented GRPO direction in the equivalent "effective-weight" form

$$\widehat{g}_{\mathrm{GRPO},i} := \frac{1}{G} \sum_{j=1}^{G} m_{i,j} g_{i,j}, \tag{22}$$

where the (random) effective multiplier $m_{i,j}$ is

$$m_{i,j} := \begin{cases} \min(\hat{\rho}_{i,j}, 1+\epsilon), & \widehat{A}_j \geq 0, \\ \max(\hat{\rho}_{i,j}, 1-\epsilon), & \widehat{A}_j < 0. \end{cases} \tag{23}$$

This form is exactly equivalent to the usual GRPO "min–clip" surrogate: for $\widehat{A}_j \geq 0$ the weight is clipped at $1+\epsilon$, while for $\widehat{A}_j < 0$ the weight is not clipped from above.

We do not assume a particular optimizer beyond the update form
$$\theta_{i+1} = \theta_i + \eta_0 \, \widehat{g}_{\mathrm{GRPO},i}$$
for some learning rate $\eta_0 > 0$.

**Filtration.** Let $\mathcal{F}_i$ denote the $\sigma$-field generated by all algorithmic randomness up to and including step $i$. Then $\theta_i$ is $\mathcal{F}_{i-1}$-measurable; hence each drift value $\Delta\mathcal{L}_{i,j}$ is $\mathcal{F}_{i-1}$-measurable.

**Standing conditions (C1–C5).** We work under the following conditions. (C1) is standard and typically enforced by gradient clipping; (C3) holds when the Monte Carlo proxy evaluations use independent randomness across samples; and (C5) is *empirically checkable* by monitoring $\sum_{j \neq j^\dagger} m_{i,j}$.

- **(C1) Bounded per-sample directions.** There exists $B < \infty$ such that $\|g_{i,j}\| \leq B$ for all inner steps $i$ and all samples $j$.

- **(C2) Conditional common right-tail envelope for log-ratio noise.** For each inner step $i$ and sample $j$, define the conditional survival function    $\bar{F}_{j,i}(z) := \mathbb{P}(\Delta\eta_{i,j} \geq z \mid \mathcal{F}_{i-1}), \qquad z \in \mathbb{R}.$
  Assume that for every $j \in \mathcal{N}$ and every $i$, $\bar{F}_{j,i}$ has unbounded support in the sense that $\bar{F}_{j,i}(z) > 0$ for all $z \in \mathbb{R}$ almost surely. Moreover, assume there exists a deterministic nonincreasing function $\bar{F} : \mathbb{R} \to (0, 1]$ (a *uniform* tail lower envelope) such that almost surely, for all $i$ and all $j \in \mathcal{N}$,
  $$\bar{F}_{j,i}(z) \geq \bar{F}(z) \qquad \forall z \in \mathbb{R}.$$

- **(C3) Conditional independence across samples.** For each inner step $i$, conditional on $\mathcal{F}_{i-1}$, the noises $\{\Delta\eta_{i,j}\}_{j=1}^{G}$ are independent. Equivalently, conditional on $\mathcal{F}_{i-1}$, the ratios $\{\hat{\rho}_{i,j}\}_{j=1}^{G}$ (and thus the effective weights $\{m_{i,j}\}$) are independent across $j$.

- **(C4) Nontrivial proxy gradient at the drift-maximizer.** There exists $b_0 > 0$ such that for all inner steps $i$, the drift-maximizer in the negative set satisfies $\|h_{i,j^\dagger}\| \geq b_0$.

- **(C5) Residual effective-weight moment control.** There exists a deterministic constant $W < \infty$ such that for all inner

steps $i$,

$$\mathbb{E}\left[\sum_{j\neq j^{\dagger}} m_{i,j} \,\Big|\, \mathcal{F}_{i-1}\right] \leq W, \qquad \text{a.s.}$$

*Remark* C.4. Condition (C5) upper-bounds the *expected total effective-weight mass* of the samples *other than the drift-maximizer*. This quantity is directly measurable in experiments as the sum $\sum_{j\neq j^{\dagger}} m_{i,j}$. Controlling this expectation guarantees via Markov's inequality that when $\hat{\rho}_{i,j^{\dagger}}$ is large, the drift-maximizer dominates the group update with a constant probability.

**Lemma C.5** (Ratio exceedance identity and drift monotonicity)**.** *Fix an inner step $i$, an index $j$, and a threshold $u > 0$. Then*

$$\mathbb{P}\big(\hat{\rho}_{i,j} \geq u \,\big|\, \mathcal{F}_{i-1}\big) = \mathbb{P}\big(\Delta\eta_{i,j} \geq \log u - \Delta\mathcal{L}_{i,j} \,\big|\, \mathcal{F}_{i-1}\big) = \bar{F}_{j,i}\big(\log u - \Delta\mathcal{L}_{i,j}\big). \tag{24}$$

*Moreover, conditional on $\mathcal{F}_{i-1}$, the map $\Delta\mathcal{L} \mapsto \bar{F}_{j,i}(\log u - \Delta\mathcal{L})$ is nondecreasing.*

*Proof.* By definition, $\hat{\rho}_{i,j} = \exp(\Delta\mathcal{L}_{i,j} + \Delta\eta_{i,j})$. Since $\exp(\cdot)$ is strictly increasing,
$$\{\hat{\rho}_{i,j} \geq u\} \iff \{\Delta\mathcal{L}_{i,j} + \Delta\eta_{i,j} \geq \log u\} \iff \{\Delta\eta_{i,j} \geq \log u - \Delta\mathcal{L}_{i,j}\}.$$
Taking conditional probabilities given $\mathcal{F}_{i-1}$ yields (24). For monotonicity, conditional on $\mathcal{F}_{i-1}$ the survival function $\bar{F}_{j,i}$ is nonincreasing in its argument, while $\Delta\mathcal{L} \mapsto \log u - \Delta\mathcal{L}$ is strictly decreasing; therefore their composition is nondecreasing. $\square$

**Lemma C.6** (Dominance from a large drift-maximizer ratio via a moment bound)**.** *Fix an inner step $i$ and let $j^{\dagger} \in \arg\max_{j\in\mathcal{N}} \Delta\mathcal{L}_{i,j}$. Fix any $\lambda \in [0,1)$ and define the (step-$i$) residual vector*

$$r_i := \frac{1}{G}\sum_{j\neq j^{\dagger}} m_{i,j}g_{i,j}.$$

*Assume Conditions (C1)–(C5). Define*

$$u_0 := \frac{2BW}{\lambda a_0 b_0}. \tag{25}$$

*Then for any $u \geq u_0$,*

$$\mathbb{P}\left(\|r_i\| \leq \lambda\frac{1}{G}u\,a_0 b_0 \,\Big|\, \mathcal{F}_{i-1}, \, \hat{\rho}_{i,j^{\dagger}} \geq u\right) \geq \frac{1}{2}, \qquad \text{a.s.} \tag{26}$$

*Moreover, on the event $\{\hat{\rho}_{i,j^{\dagger}} \geq u\}$ with $u \geq 1 + \epsilon$, since $j^{\dagger} \in \mathcal{N}$ we have $m_{i,j^{\dagger}} = \hat{\rho}_{i,j^{\dagger}}$ and thus*

$$\widehat{g}_{\mathrm{GRPO},i} = -\frac{1}{G}\hat{\rho}_{i,j^{\dagger}} |\widehat{A}_{j^{\dagger}}| h_{i,j^{\dagger}} + r_i. \tag{27}$$

*Proof.* First, by (C1),

$$\|r_i\| = \left\|\frac{1}{G}\sum_{j\neq j^{\dagger}} m_{i,j}g_{i,j}\right\| \leq \frac{1}{G}\sum_{j\neq j^{\dagger}} m_{i,j}\|g_{i,j}\| \leq \frac{B}{G}\sum_{j\neq j^{\dagger}} m_{i,j}.$$

Therefore, the event

$$\left\{\sum_{j\neq j^{\dagger}} m_{i,j} \leq \frac{\lambda}{B}u\,a_0 b_0\right\}$$

implies $\|r_i\| \leq \lambda\frac{1}{G}u a_0 b_0$.

Next, by (C3), conditional on $\mathcal{F}_{i-1}$ the collection $\{m_{i,j}\}_{j\neq j^{\dagger}}$ is independent of $\hat{\rho}_{i,j^{\dagger}}$, hence independent of the event $\{\hat{\rho}_{i,j^{\dagger}} \geq u\}$. Thus, for any threshold $t > 0$,

$$\mathbb{P}\left(\sum_{j\neq j^{\dagger}} m_{i,j} > t \,\Big|\, \mathcal{F}_{i-1}, \, \hat{\rho}_{i,j^{\dagger}} \geq u\right) = \mathbb{P}\left(\sum_{j\neq j^{\dagger}} m_{i,j} > t \,\Big|\, \mathcal{F}_{i-1}\right).$$

Applying Markov's inequality and (C5) yields

$$\mathbb{P}\left(\sum_{j\neq j^{\dagger}} m_{i,j} > t \,\Big|\, \mathcal{F}_{i-1}\right) \leq \frac{\mathbb{E}\left[\sum_{j\neq j^{\dagger}} m_{i,j} \,\big|\, \mathcal{F}_{i-1}\right]}{t} \leq \frac{W}{t}.$$

Choose $t = \frac{\lambda}{B}u a_0 b_0$. If $u \geq u_0 = \frac{2BW}{\lambda a_0 b_0}$, then $W/t \leq 1/2$, hence

$$\mathbb{P}\left(\sum_{j\neq j^{\dagger}} m_{i,j} \leq \frac{\lambda}{B}u a_0 b_0 \,\Big|\, \mathcal{F}_{i-1}, \, \hat{\rho}_{i,j^{\dagger}} \geq u\right) \geq \frac{1}{2}.$$

Combining with $\|r_i\| \leq \frac{B}{G}\sum_{j\neq j^{\dagger}} m_{i,j}$ gives (26).

Finally, on $\{\hat\rho_{i,j^\dagger} \geq u\}$ with $u \geq 1 + \epsilon$, since $j^\dagger \in \mathcal{N}$ we have $m_{i,j^\dagger} = \max(\hat\rho_{i,j^\dagger}, 1 - \epsilon) = \hat\rho_{i,j^\dagger}$. Also $g_{i,j^\dagger} = \widehat{A}_{j^\dagger} h_{i,j^\dagger} = -|\widehat{A}_{j^\dagger}| h_{i,j^\dagger}$. Substituting into (22) yields (27). $\qquad\square$

**Lemma C.7** (Dominance implies a gradient spike). *Fix an inner step $i$ and let $j^\dagger$ be as above. Assume (C1) and (C4). On the event $\{\hat\rho_{i,j^\dagger} \geq u\}$ with $u \geq 1 + \epsilon$, and on any event where*

$$\|r_i\| \leq \lambda \frac{1}{G} u\, a_0 b_0, \tag{28}$$

*we have*

$$\|\widehat{g}_{\mathrm{GRPO},i}\| \geq (1 - \lambda) \frac{1}{G} u\, a_0 b_0.$$

*In particular, if*

$$u \geq \frac{GH}{(1 - \lambda)a_0 b_0}, \tag{29}$$

*then $\|\widehat{g}_{\mathrm{GRPO},i}\| \geq H$ holds on the same event.*

*Proof.* On $\{\hat\rho_{i,j^\dagger} \geq u\}$ with $u \geq 1 + \epsilon$, Lemma C.6 gives the decomposition

$$\widehat{g}_{\mathrm{GRPO},i} = -\frac{1}{G}\hat\rho_{i,j^\dagger} |\widehat{A}_{j^\dagger}| h_{i,j^\dagger} + r_i.$$

Apply the reverse triangle inequality:

$$\|\widehat{g}_{\mathrm{GRPO},i}\| \geq \frac{1}{G}\hat\rho_{i,j^\dagger}|\widehat{A}_{j^\dagger}|\|h_{i,j^\dagger}\| - \|r_i\|.$$

Since $j^\dagger \in \mathcal{N}$ implies $|\widehat{A}_{j^\dagger}| \geq a_0$, and (C4) gives $\|h_{i,j^\dagger}\| \geq b_0$, and $\hat\rho_{i,j^\dagger} \geq u$, we obtain

$$\frac{1}{G}\hat\rho_{i,j^\dagger}|\widehat{A}_{j^\dagger}|\|h_{i,j^\dagger}\| \geq \frac{1}{G}u a_0 b_0.$$

Together with (28) this yields

$$\|\widehat{g}_{\mathrm{GRPO},i}\| \geq \frac{1}{G}u a_0 b_0 - \lambda\frac{1}{G}u a_0 b_0 = (1 - \lambda)\frac{1}{G}u a_0 b_0.$$

If (29) holds, then the right-hand side is at least $H$. $\qquad\square$

**Lemma C.8** (A drift-monotone lower bound on spike probability). *Fix a step $i$ and a spike threshold $H > 0$. Let $j^\dagger \in \arg\max_{j \in \mathcal{N}} \Delta\mathcal{L}_{i,j}$ so that $\Delta\mathcal{L}_{i,j^\dagger} = D_i$. Define $u_H$ as in (15). Assume Conditions (C1)–(C5). Then, almost surely,*

$$\mathbb{P}\big(\|\widehat{g}_{\mathrm{GRPO},i}\| \geq H \,\big|\, \mathcal{F}_{i-1}\big) \geq \frac{1}{2} \cdot \mathbb{P}\big(\hat\rho_{i,j^\dagger} \geq u_H \,\big|\, \mathcal{F}_{i-1}\big). \tag{30}$$

*Moreover, by Lemma C.5 and (C2),*

$$\mathbb{P}\big(\hat\rho_{i,j^\dagger} \geq u_H \,\big|\, \mathcal{F}_{i-1}\big) = \bar{F}_{j^\dagger,i}\big(\log u_H - D_i\big) \geq \bar{F}\big(\log u_H - D_i\big), \tag{31}$$

*and the right-hand side is nondecreasing in $D_i$.*

*Proof.* Work conditionally on $\mathcal{F}_{i-1}$. Since $u_H \geq 1 + \epsilon$ by definition, on the event $\{\hat\rho_{i,j^\dagger} \geq u_H\}$ we have $m_{i,j^\dagger} = \hat\rho_{i,j^\dagger}$. By Lemma C.6 with $u = u_H$, we have

$$\mathbb{P}\left(\|r_i\| \leq \lambda\frac{1}{G}u_H a_0 b_0 \,\Big|\, \mathcal{F}_{i-1},\, \hat\rho_{i,j^\dagger} \geq u_H\right) \geq \frac{1}{2}.$$

On the intersection of $\{\hat\rho_{i,j^\dagger} \geq u_H\}$ and $\{\|r_i\| \leq \lambda\frac{1}{G}u_H a_0 b_0\}$, Lemma C.7 implies $\|\widehat{g}_{\mathrm{GRPO},i}\| \geq H$ because $u_H \geq GH/((1-\lambda)a_0 b_0)$. Therefore,

$$\mathbb{P}\big(\|\widehat{g}_{\mathrm{GRPO},i}\| \geq H \,\big|\, \mathcal{F}_{i-1}\big) \geq \mathbb{P}\big(\hat\rho_{i,j^\dagger} \geq u_H,\, \|r_i\| \leq \lambda\tfrac{1}{G}u_H a_0 b_0 \,\big|\, \mathcal{F}_{i-1}\big)$$

$$= \mathbb{P}\big(\hat\rho_{i,j^\dagger} \geq u_H \,\big|\, \mathcal{F}_{i-1}\big) \cdot \mathbb{P}\left(\|r_i\| \leq \lambda\tfrac{1}{G}u_H a_0 b_0 \,\Big|\, \mathcal{F}_{i-1},\, \hat\rho_{i,j^\dagger} \geq u_H\right)$$

$$\geq \frac{1}{2} \cdot \mathbb{P}\big(\hat\rho_{i,j^\dagger} \geq u_H \,\big|\, \mathcal{F}_{i-1}\big),$$

which proves (30). The tail identity and lower bound (31) follow from Lemma C.5 with $\Delta\mathcal{L}_{i,j^\dagger} = D_i$, and (C2). Monotonicity in $D_i$ follows from Lemma C.5. $\qquad\square$

**Lemma C.9** (Quadratic remainder for $L$-smooth functions). *Let $f : \mathbb{R}^d \to \mathbb{R}$ be differentiable and $L$-smooth on the segment $\{\theta + t(\theta' - \theta) : t \in [0,1]\}$. Then*

$$f(\theta') \leq f(\theta) + \langle \nabla f(\theta), \theta' - \theta\rangle + \frac{L}{2}\|\theta' - \theta\|^2, \qquad f(\theta') \geq f(\theta) + \langle \nabla f(\theta), \theta' - \theta\rangle - \frac{L}{2}\|\theta' - \theta\|^2.$$

*Proof.* Let $d := \theta' - \theta$ and define the univariate function

$$\varphi(t) := f(\theta + td), \qquad t \in [0,1].$$

Since $f$ is differentiable on the segment $\{\theta + td : t \in [0,1]\}$, $\varphi$ is differentiable and
$$\varphi'(t) = \langle \nabla f(\theta + td), d \rangle .$$
By the fundamental theorem of calculus,
$$f(\theta') - f(\theta) = \varphi(1) - \varphi(0) = \int_0^1 \varphi'(t)\, dt = \int_0^1 \langle \nabla f(\theta + td), d \rangle\, dt.$$
Add and subtract $\nabla f(\theta)$ inside the inner product:
$$f(\theta') - f(\theta) = \langle \nabla f(\theta), d \rangle + \int_0^1 \langle \nabla f(\theta + td) - \nabla f(\theta), d \rangle\, dt.$$
Using Cauchy–Schwarz and $L$-smoothness (i.e., $\|\nabla f(u) - \nabla f(v)\| \le L\|u - v\|$ on the segment), for each $t \in [0,1]$ we have
$$\left| \langle \nabla f(\theta + td) - \nabla f(\theta), d \rangle \right| \le \|\nabla f(\theta + td) - \nabla f(\theta)\|\, \|d\| \le L\, t\, \|d\|^2.$$
Therefore,
$$\int_0^1 \langle \nabla f(\theta + td) - \nabla f(\theta), d \rangle\, dt \le \int_0^1 Lt\|d\|^2\, dt = \frac{L}{2}\|d\|^2,$$
which gives
$$f(\theta') \le f(\theta) + \langle \nabla f(\theta), \theta' - \theta \rangle + \frac{L}{2}\|\theta' - \theta\|^2.$$
Similarly, using $\langle \nabla f(\theta + td) - \nabla f(\theta), d \rangle \ge -Lt\|d\|^2$ yields
$$f(\theta') \ge f(\theta) + \langle \nabla f(\theta), \theta' - \theta \rangle - \frac{L}{2}\|\theta' - \theta\|^2.$$
$\square$

**Theorem C.10** (One-step decrease of $\mathcal{L}$ on a dominating sample). *Fix a step $i$ and an index $j^\star \in \mathcal{N}$. Assume that at this realized step the group update is dominated by $j^\star$ in the sense that*
$$\widehat{g}_{\mathrm{GRPO},i} = -\frac{1}{G}\, \hat{\rho}_{i,j^\star}\, |\widehat{A}_{j^\star}|\, h_{i,j^\star} + r_i^\star, \qquad \|r_i^\star\| \le \lambda\, \frac{1}{G}\, \hat{\rho}_{i,j^\star}\, |\widehat{A}_{j^\star}|\, \|h_{i,j^\star}\|. \tag{32}$$
*Define*
$$v := h_{i,j^\star} = \nabla_\theta \mathcal{L}_{\theta_i}(x_{j^\star}), \qquad \eta := \frac{\eta_0}{G}\, \hat{\rho}_{i,j^\star}\, |\widehat{A}_{j^\star}|, \qquad \delta := \eta_0 r_i^\star.$$
*Then $\theta_{i+1} = \theta_i - \eta v + \delta$ and $\|\delta\| \le \lambda \eta \|v\|$. Let $f_\star(\theta) := \mathcal{L}_\theta(x_{j^\star})$. Assume $f_\star$ is $L_\star$-smooth on the realized segment $[\theta_i, \theta_{i+1}]$. Then*
$$\mathcal{L}_{\theta_i}(x_{j^\star}) - \mathcal{L}_{\theta_{i+1}}(x_{j^\star}) \ge \eta\|v\|^2 \left( (1-\lambda) - \frac{L_\star}{2}(1+\lambda)^2 \eta \right). \tag{33}$$
*In particular, if $\eta \le \frac{1-\lambda}{L_\star(1+\lambda)^2}$, then*
$$\mathcal{L}_{\theta_i}(x_{j^\star}) - \mathcal{L}_{\theta_{i+1}}(x_{j^\star}) \ge \frac{1-\lambda}{2}\, \eta\, \|v\|^2. \tag{34}$$

*Proof.* Apply Lemma C.9 to $f_\star$ at $(\theta, \theta') = (\theta_i, \theta_{i+1})$:
$$f_\star(\theta_{i+1}) \le f_\star(\theta_i) + \langle \nabla f_\star(\theta_i), \theta_{i+1} - \theta_i \rangle + \frac{L_\star}{2}\|\theta_{i+1} - \theta_i\|^2.$$
Rearrange:
$$f_\star(\theta_i) - f_\star(\theta_{i+1}) \ge -\langle v, -\eta v + \delta \rangle - \frac{L_\star}{2}\|-\eta v + \delta\|^2 = \eta\|v\|^2 - \langle v, \delta \rangle - \frac{L_\star}{2}\|-\eta v + \delta\|^2.$$
Bound $\langle v, \delta \rangle \le \|v\|\|\delta\| \le \lambda\eta\|v\|^2$. Also $\|-\eta v + \delta\| \le \eta\|v\| + \|\delta\| \le (1+\lambda)\eta\|v\|$. Substitute to obtain (33). If $\eta \le \frac{1-\lambda}{L_\star(1+\lambda)^2}$, then the bracket is at least $(1-\lambda)/2$, yielding (34). $\square$

**Definition C.11** (Anti-alignment and directional curvature). Fix distinct indices $j^\star \ne j^\diamond$ and define $x^\star := x_{j^\star}$ and $x^\diamond := x_{j^\diamond}$. Let
$$v := \nabla_\theta \mathcal{L}_{\theta_i}(x^\star), \qquad u := \nabla_\theta \mathcal{L}_{\theta_i}(x^\diamond), \qquad \gamma := -\langle u, v \rangle.$$

**Theorem C.12** (Cross-sample amplification with residual (proxy drift increase)). *Fix a step $i$ and two indices $j^\star \ne j^\diamond$ in $\mathcal{N}$. Assume the outlier-dominance decomposition (32) holds on this realized step, and define*
$$v = \nabla_\theta \mathcal{L}_{\theta_i}(x_{j^\star}), \qquad u = \nabla_\theta \mathcal{L}_{\theta_i}(x_{j^\diamond}), \qquad \gamma = -\langle u, v \rangle > 0,$$
$$\eta = \frac{\eta_0}{G}\hat{\rho}_{i,j^\star}\, |\widehat{A}_{j^\star}|, \qquad \theta_{i+1} = \theta_i - \eta v + \delta, \qquad \|\delta\| \le \lambda\eta\|v\|.$$
*Let $f_\diamond(\theta) := \mathcal{L}_\theta(x_{j^\diamond})$ and assume $f_\diamond$ is $L_\diamond$-smooth on the realized segment $[\theta_i, \theta_{i+1}]$. If*
$$\eta \le \frac{\gamma}{(1+\lambda)^2\, L_\diamond\|v\|^2}, \tag{35}$$
*then on this realized step we have*
$$\mathcal{L}_{\theta_{i+1}}(x_{j^\diamond}) - \mathcal{L}_{\theta_i}(x_{j^\diamond}) \ge \eta\Big(\frac{\gamma}{2} - \lambda\|u\|\|v\|\Big), \tag{36}$$
*and consequently,*
$$\Delta\mathcal{L}_{i+1,j^\diamond} \ge \Delta\mathcal{L}_{i,j^\diamond} + \eta\Big(\frac{\gamma}{2} - \lambda\|u\|\|v\|\Big). \tag{37}$$

*Proof.* Apply Lemma C.9 (lower bound form) to $f_\diamond$ at $(\theta, \theta') = (\theta_i, \theta_{i+1})$:

$$f_\diamond(\theta_{i+1}) \geq f_\diamond(\theta_i) + \langle \nabla f_\diamond(\theta_i), \theta_{i+1} - \theta_i \rangle - \frac{L_\diamond}{2} \|\theta_{i+1} - \theta_i\|^2.$$

Substitute $\nabla f_\diamond(\theta_i) = u$ and $\theta_{i+1} - \theta_i = -\eta v + \delta$:

$$f_\diamond(\theta_{i+1}) - f_\diamond(\theta_i) \geq \langle u, -\eta v + \delta \rangle - \frac{L_\diamond}{2} \| - \eta v + \delta \|^2 = \eta \gamma + \langle u, \delta \rangle - \frac{L_\diamond}{2} \| - \eta v + \delta \|^2.$$

Use $\langle u, \delta \rangle \geq -\|u\|\|\delta\| \geq -\lambda \eta \|u\|\|v\|$ and $\| - \eta v + \delta \| \leq (1 + \lambda)\eta \|v\|$ to get

$$f_\diamond(\theta_{i+1}) - f_\diamond(\theta_i) \geq \eta \gamma - \lambda \eta \|u\|\|v\| - \frac{L_\diamond}{2}(1 + \lambda)^2 \eta^2 \|v\|^2.$$

Under (35), the quadratic term is at most $\frac{1}{2}\eta\gamma$, yielding (36). Equation (37) is just rewriting in terms of $\Delta\mathcal{L}$. $\square$

**Lemma C.13** (From amplification of one sample to an increase in $D_i$). *Fix a step $i$ and suppose Theorem C.12 applies for some $j^\star \neq j^\diamond$ in $\mathcal{N}$. Define*

$$\eta = \frac{\eta_0}{G} \hat{\rho}_{i,j^\star} |\widehat{A}_{j^\star}|.$$

*Define the $\mathcal{F}_{i-1}$-measurable coefficient*

$$c_{\mathrm{amp},i} := \frac{\eta_0 |\widehat{A}_{j^\star}|}{G}\left( \frac{\gamma}{2} - \lambda \|u\|\|v\| \right) \in \mathbb{R},$$

*where $u = \nabla_\theta \mathcal{L}_{\theta_i}(x_{j^\diamond})$, $v = \nabla_\theta \mathcal{L}_{\theta_i}(x_{j^\star})$, and $\gamma = -\langle u, v \rangle > 0$. Then on this realized step,*

$$D_{i+1} \geq D_i + \left( c_{\mathrm{amp},i} \hat{\rho}_{i,j^\star} - S_i \right). \tag{38}$$

*Proof.* By Theorem C.12,

$$\Delta\mathcal{L}_{i+1,j^\diamond} \geq \Delta\mathcal{L}_{i,j^\diamond} + \eta\left( \frac{\gamma}{2} - \lambda \|u\|\|v\| \right) = \Delta\mathcal{L}_{i,j^\diamond} + c_{\mathrm{amp},i} \hat{\rho}_{i,j^\star}.$$

Since $D_{i+1} = \max_{j \in \mathcal{N}} \Delta\mathcal{L}_{i+1,j} \geq \Delta\mathcal{L}_{i+1,j^\diamond}$, we have

$$D_{i+1} \geq \Delta\mathcal{L}_{i,j^\diamond} + c_{\mathrm{amp},i} \hat{\rho}_{i,j^\star}.$$

By definition of $S_i$ in (14),

$$\Delta\mathcal{L}_{i,j^\diamond} \geq \min_{j \in \mathcal{N}} \Delta\mathcal{L}_{i,j} = D_i - S_i.$$

Substituting yields (38). $\square$

*Proof of Theorem C.1.* The spike-probability bound (17) follows from Lemma C.8 and (C2):

$$\mathbb{P}\left( \|\widehat{g}_{\mathrm{GRPO},i}\| \geq H \,\big|\, \mathcal{F}_{i-1} \right) \geq \frac{1}{2}\bar{F}(\log u_H - D_i).$$

Monotonicity in $D_i$ holds by Lemma C.5.

For the one-step decrease bound on a dominating sample, consider a realized step $i$ where a negative-advantage sample $j^\star \in \mathcal{N}$ with $\hat{\rho}_{i,j^\star} \geq 1 + \epsilon$ dominates the group update in the sense of (32) and where the local smoothness/step-size condition of Theorem C.10 holds, including (34). Then (34) gives

$$\mathcal{L}_{\theta_i}(x_{j^\star}) - \mathcal{L}_{\theta_{i+1}}(x_{j^\star}) \geq \frac{1 - \lambda}{2} \eta \|v\|^2 = \frac{1 - \lambda}{2} \frac{\eta_0}{G} \hat{\rho}_{i,j^\star} |\widehat{A}_{j^\star}| \|v\|^2,$$

with $v = \nabla_\theta \mathcal{L}_{\theta_i}(x_{j^\star})$. Thus the first inequality in (18) holds with

$$c_{\mathrm{sup},i} := \frac{1 - \lambda}{2} \eta_0 |\widehat{A}_{j^\star}| \|v\|^2 > 0.$$

For the drift-state increment, Lemma C.13 gives

$$D_{i+1} \geq D_i + \left( c_{\mathrm{amp},i} \hat{\rho}_{i,j^\star} - S_i \right),$$

establishing the second inequality in (18).

Finally, if $c_{\mathrm{amp},i} \hat{\rho}_{i,j^\star} \geq S_i$ then $D_{i+1} \geq D_i$. Since $D \mapsto \bar{F}(\log u_H - D)$ is nondecreasing (Lemma C.5), the lower bound $P_i(H) = \frac{1}{2}\bar{F}(\log u_H - D_i)$ cannot decrease from step $i$ to step $i+1$ on that realized step, i.e., (19) holds. $\square$

## C.3 Proof of Theorem C.2

We prove Theorem C.2 for the unconditional upper-clipping rule $w_{i,j} = \min(\hat{\rho}_{i,j}, 1 + \epsilon)$ and $\widehat{g}_{\mathrm{clip},i} := \frac{1}{G} \sum_{j=1}^{G} w_{i,j} g_{i,j}$.

**Lemma C.14** (A sufficient upper-bound dominance event under upper clipping). *Fix an inner step $i$ and let $j^\dagger \in \arg\max_{j \in \mathcal{N}} \Delta\mathcal{L}_{i,j}$. Assume (C1) and (C4). Define the residual*

$$r_i^{\mathrm{clip}} := \frac{1}{G} \sum_{j \neq j^\dagger} w_{i,j} g_{i,j}.$$

*On any realized step where $\hat{\rho}_{i,j^\dagger} \geq 1 + \epsilon$ and*

$$\sum_{j \neq j^\dagger} w_{i,j} \leq \frac{\lambda a_0 b_0}{B}(1 + \epsilon), \tag{39}$$

*we have the deterministic decomposition*

$$\widehat{g}_{\mathrm{clip},i} = -\frac{1}{G}(1 + \epsilon)\,|\widehat{A}_{j^\dagger}|\,h_{i,j^\dagger} + r_i^{\mathrm{clip}}, \qquad \|r_i^{\mathrm{clip}}\| \leq \lambda \frac{1}{G}(1 + \epsilon)\,|\widehat{A}_{j^\dagger}|\,\|h_{i,j^\dagger}\|.$$

*Proof.* On $\hat{\rho}_{i,j^\dagger} \geq 1 + \epsilon$, we have $w_{i,j^\dagger} = 1 + \epsilon$ and $g_{i,j^\dagger} = \widehat{A}_{j^\dagger} h_{i,j^\dagger} = -|\widehat{A}_{j^\dagger}|h_{i,j^\dagger}$. Thus

$$\widehat{g}_{\mathrm{clip},i} = \frac{1}{G} w_{i,j^\dagger} g_{i,j^\dagger} + \frac{1}{G} \sum_{j \neq j^\dagger} w_{i,j} g_{i,j} = -\frac{1}{G}(1 + \epsilon)|\widehat{A}_{j^\dagger}|h_{i,j^\dagger} + r_i^{\mathrm{clip}}.$$

Moreover, by (C1),

$$\|r_i^{\mathrm{clip}}\| \leq \frac{1}{G} \sum_{j \neq j^\dagger} w_{i,j}\|g_{i,j}\| \leq \frac{B}{G} \sum_{j \neq j^\dagger} w_{i,j}.$$

Under (39), this yields

$$\|r_i^{\mathrm{clip}}\| \leq \frac{B}{G} \cdot \frac{\lambda a_0 b_0}{B}(1 + \epsilon) = \lambda \frac{1}{G}(1 + \epsilon)a_0 b_0 \leq \lambda \frac{1}{G}(1 + \epsilon)|\widehat{A}_{j^\dagger}|\|h_{i,j^\dagger}\|,$$

since $|\widehat{A}_{j^\dagger}| \geq a_0$ and $\|h_{i,j^\dagger}\| \geq b_0$ by (C4). $\qquad\square$

*Proof of Theorem C.2.* Fix an inner step $i$. By (C1) we have $\|g_{i,j}\| \leq B$ for all $j$. Moreover, since $\hat{\rho}_{i,j} > 0$ and $w_{i,j} = \min(\hat{\rho}_{i,j}, 1 + \epsilon)$, we have $0 < w_{i,j} \leq 1 + \epsilon$. Therefore,

$$\|\widehat{g}_{\mathrm{clip},i}\| = \left\|\frac{1}{G} \sum_{j=1}^{G} w_{i,j}\, g_{i,j}\right\| \leq \frac{1}{G} \sum_{j=1}^{G} w_{i,j}\,\|g_{i,j}\| \leq (1 + \epsilon)B,$$

which proves the deterministic boundedness claim.

Let $j^\dagger \in \arg\max_{j \in \mathcal{N}} \Delta\mathcal{L}_{i,j}$ so that $\Delta\mathcal{L}_{i,j^\dagger} = D_i$. By Lemma C.5 with $u = 1 + \epsilon$ and (C2),
$$\mathbb{P}(\hat{\rho}_{i,j^\dagger} \geq 1 + \epsilon \,|\, \mathcal{F}_{i-1}) = \bar{F}_{j^\dagger,i}(\log(1 + \epsilon) - D_i) \geq \bar{F}(\log(1 + \epsilon) - D_i),$$
and the right-hand side is nondecreasing in $D_i$ by Lemma C.5, which establishes (20).

Finally, on any realized step where the sufficient dominance event in Lemma C.14 holds and where the local smoothness/geometry conditions required by Theorem C.12 (with the effective step size $\eta = \frac{\eta_0}{G}(1 + \epsilon)|\widehat{A}_{j^\dagger}|$) hold for some $j^\diamond \in \mathcal{N} \setminus \{j^\dagger\}$, the same argument as Lemma C.13 yields
$$D_{i+1} \geq D_i + \big(c_{\mathrm{amp},i}(1 + \epsilon) - S_i\big).$$
This completes the proof. $\qquad\square$

## C.4  Proof of Theorem C.3

*Proof of Theorem C.3.* Fix an inner step $i$ and define the upper-clipped positive weights
$$w_{i,j} := \mathrm{clip}_\epsilon^+(\hat{\rho}_{i,j}) = \min(\hat{\rho}_{i,j}, 1 + \epsilon), \qquad j = 1, \ldots, G.$$
Since $\hat{\rho}_{i,j} > 0$, we have $w_{i,j} > 0$ and thus $\sum_{k=1}^{G} w_{i,k} > 0$. Define
$$\widehat{g}_{\mathrm{sn},i} := \frac{\sum_{j=1}^{G} w_{i,j}\, g_{i,j}}{\sum_{j=1}^{G} w_{i,j}}.$$
Let $\alpha_{i,j} := w_{i,j} / \sum_{k=1}^{G} w_{i,k}$. Then $\alpha_{i,j} \geq 0$ and $\sum_{j=1}^{G} \alpha_{i,j} = 1$, hence
$$\widehat{g}_{\mathrm{sn},i} = \sum_{j=1}^{G} \alpha_{i,j} g_{i,j} \in \mathrm{conv}\{g_{i,1}, \ldots, g_{i,G}\}.$$
By (C1), $\|g_{i,j}\| \leq B$ for all $j$, therefore
$$\|\widehat{g}_{\mathrm{sn},i}\| \leq \sum_{j=1}^{G} \alpha_{i,j}\|g_{i,j}\| \leq \sum_{j=1}^{G} \alpha_{i,j} B = B.$$
This proves the deterministic bound and the convex-hull property. $\qquad\square$

## C.5   Bias from clipped self-normalization

**Lemma C.15** (Coordinate-wise bias of clipped self-normalization). *Consider one rollout group and define the coordinate-wise per-sample contribution*

$$z_{j,r} := \left(\widehat{A}_j \nabla_\theta \mathcal{L}_\theta(x_j)\right)_r, \qquad |z_{j,r}| \le B_r.$$

*Let*

$$\bar{w}_j := \min(\widehat{\rho}_j, 1+\epsilon), \qquad M := \frac{1}{G} \sum_{j=1}^{G} \bar{w}_j.$$

*Define the standard importance-weighted coordinate update and the clipped self-normalized coordinate update by*

$$U_{\mathrm{iw},r} := \frac{1}{G} \sum_{j=1}^{G} \widehat{\rho}_j z_{j,r}, \qquad U_{\mathrm{sn},r} := \frac{\sum_{j=1}^{G} \bar{w}_j z_{j,r}}{\sum_{j=1}^{G} \bar{w}_j}.$$

*Then the coordinate-wise bias*

$$\Delta_r := \mathbb{E}[U_{\mathrm{sn},r} - U_{\mathrm{iw},r}]$$

*satisfies*

$$|\Delta_r| \le B_r \, \mathbb{E}\left[\left|1 - \frac{1}{G} \sum_{j=1}^{G} \bar{w}_j\right|\right] + \frac{B_r}{G} \, \mathbb{E}\left[\sum_{j=1}^{G} (\widehat{\rho}_j - (1+\epsilon))_+\right].$$

*The first term measures the deviation of the clipped normalizer from one, and the second term measures the removed upper-tail mass.*

*Proof.* Let

$$C_r := \frac{1}{G} \sum_{j=1}^{G} \bar{w}_j z_{j,r}, \qquad T_r := \frac{1}{G} \sum_{j=1}^{G} (\widehat{\rho}_j - \bar{w}_j) z_{j,r}.$$

Then

$$U_{\mathrm{iw},r} = C_r + T_r, \qquad U_{\mathrm{sn},r} = \frac{C_r}{M},$$

because $\sum_{j=1}^{G} \bar{w}_j = GM$. Therefore,

$$U_{\mathrm{sn},r} - U_{\mathrm{iw},r} = C_r\left(\frac{1}{M} - 1\right) - T_r.$$

Since $\bar{w}_j > 0$ and $|z_{j,r}| \le B_r$,

$$|C_r| \le \frac{1}{G} \sum_{j=1}^{G} \bar{w}_j B_r = B_r M.$$

Hence

$$\left|C_r\left(\frac{1}{M} - 1\right)\right| \le B_r |1 - M|.$$

Moreover,

$$\widehat{\rho}_j - \bar{w}_j = \widehat{\rho}_j - \min(\widehat{\rho}_j, 1+\epsilon) = (\widehat{\rho}_j - (1+\epsilon))_+,$$

so

$$|T_r| \le \frac{B_r}{G} \sum_{j=1}^{G} (\widehat{\rho}_j - (1+\epsilon))_+.$$

Combining the two bounds gives

$$|U_{\mathrm{sn},r} - U_{\mathrm{iw},r}| \le B_r |1 - M| + \frac{B_r}{G} \sum_{j=1}^{G} (\widehat{\rho}_j - (1+\epsilon))_+.$$

Taking expectations and using $|\mathbb{E}[X]| \le \mathbb{E}[|X|]$ proves the claim. ☐

# D Experimental Details

## D.1 Training and Hyperparameter Setup

We provide detailed configurations for our experiments on both Full-Attention Diffusion and Block Diffusion architectures to ensure reproducibility. All experiments were conducted using the StableDRL framework, with hyperparameters chosen to isolate the contribution of our stability mechanisms.

### D.1.1 FULL-ATTENTION DIFFUSION (LLADA-8B-INSTRUCT)

We fine-tune the `LLaDA-8B-Instruct` model using iterative decoding with a generation length of 256 tokens and a block size of 32. Optimization is performed using AdamW with a learning rate of $1.0 \times 10^{-6}$ and a linear decay schedule over 2,000 steps. Crucially, we enable Self-Normalized Importance Sampling (SNIS) with $\epsilon = 5$, corresponding to an upper weight cap of $1 + \epsilon$.

Table 6 summarizes the complete hyperparameter configuration.

*Table 6.* Hyperparameter Configuration for Full-Attention Diffusion (LLaDA-8B-Instruct)

| Category | Value |
|---|---|
| *Model & Initialization* | |
| Base Model | LLaDA-8B-Instruct |
| Precision | bfloat16 |
| Activation Checkpointing | Whole Layer |
| *Generation (Rollout)* | |
| Decoding Strategy | Iterative (128 steps) |
| Generation Length | 256 tokens |
| Block Size | 32 |
| Temperature | 0.9 |
| Rollout Scale | 8 generations $\times$ 2 repeats |
| *Training & Optimization* | |
| Optimizer | AdamW ($\beta_1 = 0.9, \beta_2 = 0.99, \lambda = 0.1$) |
| Learning Rate | $1.0 \times 10^{-6}$ (Linear Decay) |
| Batch Size | 1 (Grad Accumulation = 4) |
| Gradient Clipping | 0.2 |
| Inner Updates | 2 per rollout cycle |
| Total Steps | 2000 |
| *StableDRL Specifics* | |
| Loss Function | Sandwiched ($\beta = 1.5, \omega = 0.5$) |
| ELBO Estimation | 2 MC samples (perturbation $p = 0.15$) |
| Stabilization | SN enabled; $\epsilon = 5$; cap $1 + \epsilon$ |

The matched GSM8K full-fine-tuning and LoRA reruns are reported in Table 5 in the main text.

### D.1.2 BLOCK DIFFUSION (SDAR-8B-CHAT)

We instantiate StableDRL on the `SDAR-8B-Chat` architecture, following the conventions of TraceRL extended with our stability mechanisms. We utilize dynamic sampling with a threshold of $\tau = 0.9$ and a temperature of 1.0. The model is trained using AdamW with a learning rate of $1.0 \times 10^{-6}$ and no weight decay. To stabilize the group-wise updates, we employ Group-wise SNIS with $\epsilon = 5$, implemented as an upper log-ratio cap $\log(1 + \epsilon)$. We also enable mask resampling in the trainer to maintain valid drift coupling during optimization.

Table 7 details the configuration for the block diffusion experiments.

*Table 7.* Hyperparameter Configuration for Block Diffusion (SDAR-8B-Chat)

| Category | Value |
|---|---|
| *Model & Initialization* | |
| Base Model | JetLM/SDAR-8B-Chat |
| Architecture | Block Diffusion ($B = 4$) |
| Precision | bf16 (TF32 enabled) |
| *Generation (Rollout)* | |
| Sampling Strategy | Dynamic ($\tau = 0.9, T = 1.0$) |
| Denoising Steps | 4 per block |
| Rollout Scale | 16 responses per task |
| *Training & Optimization* | |
| Optimizer | AdamW ($lr = $ 1e-6, $\beta_2 = 0.999$, no decay) |
| Scheduler | Linear Decay |
| Micro Batch Size | 1 (Gradient Accumulation = 2) |
| Gradient Clipping | 1.0 |
| *StableDRL Specifics* | |
| Advantage Mode | Raw Centered |
| Importance Sampling | Group-wise SNI |
| Clip Threshold | $\epsilon = 5$; log cap $\log(1 + \epsilon)$ |
| Mask Resampling | Enabled |

## D.2  Details of the Exploding Importance Weight Protocol

To validate the robustness of StableDRL against the heavy-tailed noise characteristic of dLLMs, we use a controlled adversarial protocol that artificially inflates the variance of the importance ratio $\hat{\rho}$. As a direct noise-assumption diagnostic, we estimate ratio-proxy noise from 8 GSM8K rollouts using a 1000-sample MC oracle ELBO and bootstrap subset sizes $m \in \{2, 5, 10\}$.

The direct noise-assumption validation results are reported in Table 3 in the main text.

## D.3  Mechanism: Asymmetric Masking

The importance ratio is estimated as $\hat{\rho} = \exp(\hat{\mathcal{L}}_\theta - \hat{\mathcal{L}}_{\text{old}})$. We induce "exploding" weights by breaking the symmetry of the Monte Carlo estimation for a random 70% subset of the batch (the "stressed" samples). We employ two decoupled masking policies:

1. **Numerator ($\hat{\mathcal{L}}_\theta$) → "Easy" Masking:** We bias masking towards high-confidence regions (e.g., the sequence tail) and select the *minimum* number of masked tokens ($t_{\min}$). This yields a tighter, optimistic ELBO estimate.

2. **Denominator ($\hat{\mathcal{L}}_{\text{old}}$) → "Hard" Masking:** We bias masking towards low-confidence regions (e.g., the sequence head) and select the *maximum* number of masked tokens ($t_{\max}$). This yields a looser, pessimistic ELBO estimate.

This systematic gap ensures that $\hat{\mathcal{L}}_\theta \gg \hat{\mathcal{L}}_{\text{old}}$, driving $\hat{\rho} \to \infty$ purely due to estimation variance, independent of the actual policy probability.

## D.4  Implementation

We operationalize "Easy" vs. "Hard" based on the diffusion formulation (Block vs. Random Token). Algorithm 1 details the generation process.

## D.5  Stress Testing Exploding Importance Ratios

A central hypothesis of this work is that dLLM training instability is driven by heavy-tailed importance weights ($\hat{\rho}$) derived from stochastic ELBO proxies. To rigorously isolate this factor, we design an adversarial *Exploding Weight Stress Test*

---

**Algorithm 1** Adversarial Generation of Exploding Importance Weights

---

**Require:** Batch $X$, Group size $G$, Coverage fraction $\gamma = 0.7$
**Require:** Bias Strength $\beta = 6.0$ (for Random), Masking Policy $\mathcal{P} \in \{\text{Block}, \text{Random}\}$
 1: **for** each group $g$ in Batch **do**
 2:     Select indices $S_g \subset g$ with size $\lceil \gamma \cdot G \rceil$ to stress.
 3:     **for** each sample $x_i$ in group $g$ **do**
 4:       **if** $i \in S_g$ **then**
 5:          // 1. Numerator: "Easy" Masking (Tail Bias + Min Count)
 6:          **if** $\mathcal{P}$ is Block **then**
 7:             $M_{\text{num}} \leftarrow$ Mask indices of the **Last Block** (Max Context)
 8:          **else**
 9:             $W[k] \propto \exp(+\beta \cdot k/L)$ {Tail Position Bias}
10:             $M_{\text{num}} \sim \text{Multinomial}(W)$
11:             $\text{Count}(M_{\text{num}}) \leftarrow t_{\min}$ {Min Masked Tokens}
12:          **end if**
13:          // 2. Denominator: "Hard" Masking (Head Bias + Max Count)
14:          **if** $\mathcal{P}$ is Block **then**
15:             $M_{\text{den}} \leftarrow$ Mask indices of the **First Block** (Min Context)
16:          **else**
17:             $W[k] \propto \exp(-\beta \cdot k/L)$ {Head Position Bias}
18:             $M_{\text{den}} \sim \text{Multinomial}(W)$
19:             $\text{Count}(M_{\text{den}}) \leftarrow t_{\max}$ {Max Masked Tokens}
20:          **end if**
21:       **else**
22:          // Control: Standard Uniform Masking
23:          $M_{\text{num}}, M_{\text{den}} \sim \text{UniformRandom}(x_i)$
24:       **end if**
25:       $\hat{\mathcal{L}}_\theta \leftarrow \text{ComputeELBO}(x_i, \pi_\theta, M_{\text{num}})$
26:       $\hat{\mathcal{L}}_{\text{old}} \leftarrow \text{ComputeELBO}(x_i, \pi_{\text{old}}, M_{\text{den}})$
27:       $\hat{\rho}_i \leftarrow \exp(\hat{\mathcal{L}}_\theta - \hat{\mathcal{L}}_{\text{old}})$
28:     **end for**
29: **end for**
30: **return** Importance Weights $\hat{\rho}$

---

(protocol details in Appendix D.2). This protocol synthesizes "Exploding" weights for a subset of trajectories by pairing "easy" masking patterns (high ELBO) with "hard" masking patterns (low ELBO), artificially amplifying proxy variance without altering the ground-truth data or rewards.

Figure 6 compares StableDRL, SPG, and ESPO under both "Normal" (unbiased) and "Exploding" conditions.

**StableDRL (Ours): Invariant Stability.** StableDRL demonstrates remarkable robustness. In the Normal setting (Green Solid), it achieves the highest final reward. Crucially, under the Exploding setting (Green Dashed), although the extreme noise in the importance weight has a minor impact on the performance, the training remains stable and monotonic.

**ESPO: Noise-Accelerated Collapse.** ESPO exhibits a clear sensitivity to noise magnitude. In the Normal setting (Orange Solid), it learns transiently before collapsing around step 500. However, in the Exploding setting (Orange Dashed), the collapse is immediate and catastrophic ($<200$ steps). This confirms that the conditional clipping of the original GRPO (Sec. 3.1) is the primary failure mechanism: when proxy noise increases, the frequency of negative-advantage outliers bypassing the clip boundary rises, destroying the policy.

**SPG: Bias-Induced Failure.** SPG (Blue) collapses in *both* settings. Because SPG reuses rollouts without importance sampling correction (implicitly assuming $\rho = 1$), it is technically immune to weight explosions. However, this immunity comes at the cost of uncontrolled off-policy bias. As the policy drifts, the discrepancy between the behavior and target distributions accumulates, leading to degradation regardless of the proxy noise level. This result underscores that simply ignoring importance weights (to avoid variance) is not a viable strategy for long-horizon alignment.

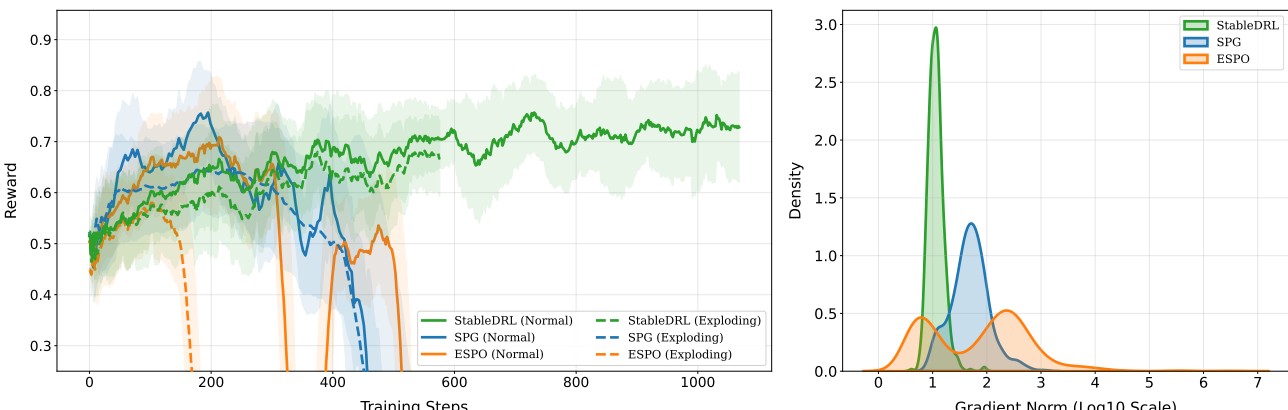

*Figure 6.* **Robustness to Proxy Noise: The "Exploding Weight" Stress Test (GSM8K).** We compare training stability under standard conditions ("Normal", solid lines) versus an adversarial regime where importance weight variance is artificially amplified ("Exploding", dashed lines; see App. D.2). **(Left) Reward Trajectories:** StableDRL (Green) demonstrates *invariant stability*, maintaining monotonic improvement in both regimes. In contrast, ESPO (Orange) suffers immediate, **noise-accelerated collapse**, confirming its sensitivity to ratio outliers. SPG (Blue) degrades in both settings, indicating that avoiding ratios (to reduce variance) fatally exposes the model to off-policy bias. **(Right) Gradient Norm Density:** Visualizing the failure mechanism. StableDRL maintains a condensed, low-variance gradient distribution. Conversely, ESPO exhibits a heavy right tail of explosive updates (log-norm > 3), confirming that the "Asymmetric Clipping Failure" allows noise spikes to propagate unchecked.

The same-task Countdown collapse consistency statistics are reported in Table 4 in the main text.

### D.6 Instability Mechanism Curves

To bridge the gap between the instability analysis in Sec. 3.1 and observed training dynamics, we perform a fine-grained analysis of gradient-norm evolution. We quantify instability using the *relative gradient spike rate*. A step $t$ is classified as a spike if its gradient norm exceeds the local moving average by a margin $\delta$:

$$\mathbb{I}_{\text{spike}}^{(t)} = \mathbb{1}\left[\|g_t\| > (1 + \delta) \cdot \frac{1}{W} \sum_{k=1}^{W} \|g_{t-k}\|\right], \tag{40}$$

where we use window $W = 50$ and margin $\delta = 0.3$. Figure 7 reports the reward trajectory, relative spike rate, and the next-step spike threshold $(1 + \delta)\frac{1}{W} \sum_{k=1}^{W} \|g_{t-k}\|$ for GRPO/ESPO, unconditional upper clipping, and StableDRL. We use ESPO (Ou et al., 2025a) as the representative GRPO-style baseline and implement unconditional upper clipping in our training framework with the same clipping threshold $\epsilon$ and optimization settings. All curves correspond to full-parameter RL fine-tuning and are intended as mechanism diagnostics; the same-task Countdown statistics in Table 4 provide the primary matched collapse evidence.

As shown in Figure 7, GRPO/ESPO exhibits divergent behavior: unbounded estimated ratios drive an increasing spike rate and eventual reward collapse. Unconditional upper clipping bounds individual ratio magnitudes, but without self-normalization the clipped group mass can still fluctuate substantially, producing a high-frequency boundary-saturated spike regime. These bounded but frequent shocks distort the optimizer history and destabilize learning. StableDRL combines upper clipping with self-normalization, keeping the spike rate low and stable while decoupling update magnitude from clipped group-mass fluctuations; this yields smooth, monotonic reward improvement.

### D.7 Visual Diagnosis of Gradient Instability

To empirically validate the "Instability Feedback Loop" and the structural failures diagnosed in Section 3.1, we visualize the joint distribution of importance weights ($\log_{10} \rho$) and gradient norms ($\log_{10} \|\hat{g}\|$) recorded during training. Figure 8 presents a comparative diagnostic of ESPO, SPG-IS, and StableDRL, offering a direct geometric validation of our theoretical analysis.

**The "Chimney" Failure in ESPO.** As observed in the left panel, ESPO exhibits a pathological "chimney" distribution. While the majority of samples cluster in a low-variance region, a sparse subset of noise-induced outliers (importance weights $\rho > 10^6$) drives gradient norms to catastrophic levels ($\|\hat{g}\| > 10^4$). This empirically confirms *Failure 1 (Asymmetric Failure of the Clipped Surrogate)* described in Section 3.1: when a sample with a large noise-induced importance weight has a

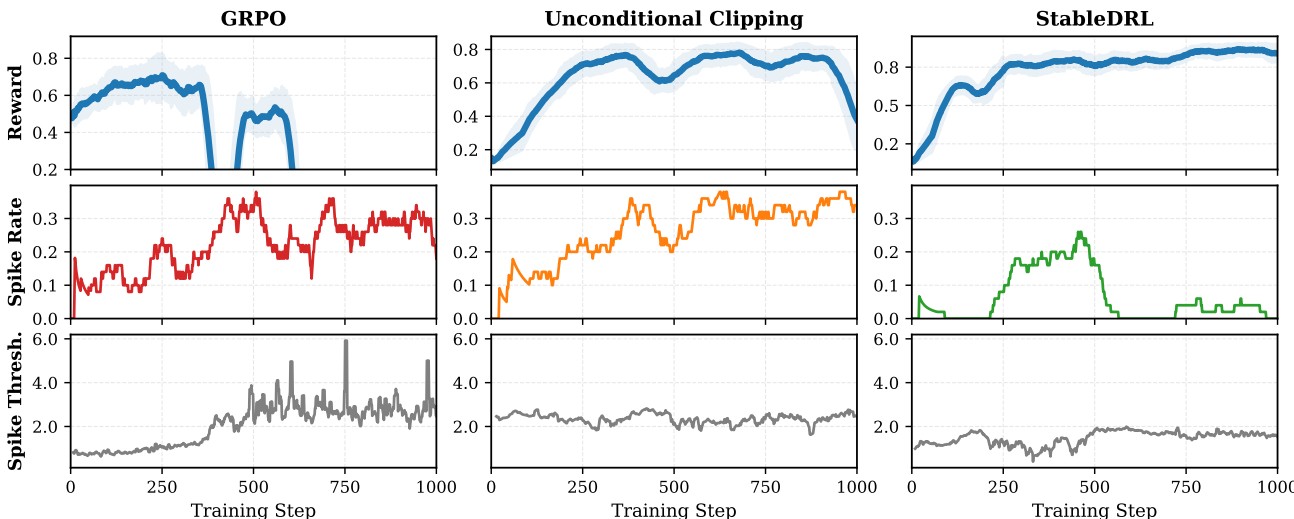

*Figure 7.* **Verification of instability mechanisms across methods. Left Column (GRPO/ESPO):** Unbounded estimated ratios fuel an accelerating spike rate and reward collapse. **Middle Column (Unconditional Upper Clipping):** Upper clipping bounds individual ratios but can produce a high-frequency, boundary-saturated spike regime that destabilizes learning. **Right Column (StableDRL):** StableDRL maintains a low, stable spike rate decoupled from clipped group-mass fluctuations, resulting in monotonic reward improvement. The diagnostic curves are complemented by the same-task Countdown consistency check in Table 4.

negative advantage, it falls into the unclipped branch of the objective. Consequently, these "trapdoor" outliers bypass the trust region and act as unbounded multipliers on the step size, injecting massive shocks that destabilize the policy.

**Drift-Variance Correlation in SPG-IS.** The center panel displays the dynamics of SPG-IS. Although SPG avoids explicit ratio computation to mitigate the "chimney" effect, the visualization reveals a strong positive correlation between the implicit weight magnitude and the update norm. This indicates that the method remains sensitive to policy drift: as the target policy diverges from the behavior policy, the accumulated "rollout-reuse bias" scales the variance of the updates proportionally. This prevents convergence, as the method lacks the structural constraints to mechanically decouple the update magnitude from distribution shifts.

**Geometric Stability in StableDRL.** In contrast, the right panel demonstrates the efficacy of our proposed framework. StableDRL displays a compact, bounded distribution where gradient norms remain consistently low ($< 10^{1.8}$) regardless of the importance weight magnitude. This confirms the effect of our dual stability mechanisms: *Unconditional Clipping* strictly censors extreme ratios before aggregation, while *Self-Normalization* ensures the update remains a convex combination of per-sample advantage-weighted update directions. As predicted by *Theorem 3.5*, StableDRL effectively confines the update to the convex hull of the per-sample advantage-weighted directions, maintaining deterministic stability even in the presence of heavy-tailed proxy noise.

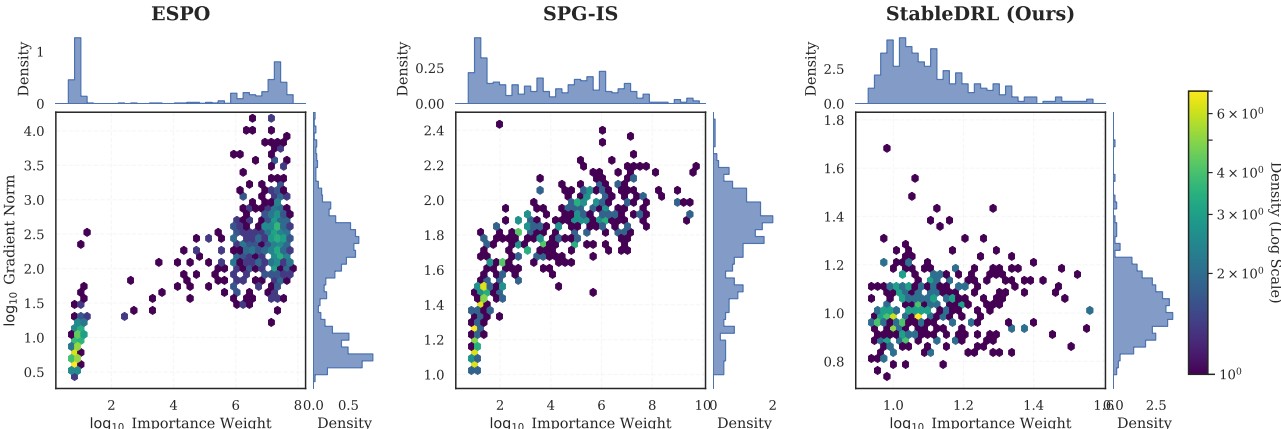

*Figure 8.* **Diagnosing Gradient Instability in dLLM Training.** We visualize the joint distribution of importance weights ($\log_{10} \rho$) and gradient norms ($\log_{10} \|\hat{g}\|$) during training. **(Left) ESPO:** Exhibits a characteristic "chimney" failure where rare, noise-induced outliers bypass clipping on negative advantages, acting as unbounded step-size multipliers that drive gradients to explosion ($> 10^4$). **(Center) SPG-IS:** Despite avoiding explicit ratios, the update variance is strongly correlated with policy drift, confirming that rollout-reuse bias accumulates to destabilize training. **(Right) StableDRL (Ours):** By enforcing strict clipping and self-normalization, our method decouples update magnitude from proxy noise, confining updates to the convex hull of the per-sample advantage-weighted directions (Theorem 3.5) and maintaining deterministic stability.

