# OpenReview forum: "Stabilizing Reinforcement Learning for Diffusion Language Models"
_ICML.cc/2026/Conference — ICML 2026 regular_

### Official Review · Reviewer_2P3A · 2026-03-10

**Soundness:** 3
**Presentation:** 3
**Significance:** 2
**Originality:** 3
**Overall Recommendation:** 4
**Confidence:** 4

**Summary:**

This paper studies why GRPO-style reinforcement learning is unstable when applied to discrete diffusion LLMs (dLLMs) whose importance ratios must be estimated from noisy likelihood proxies. The authors identify a self-reinforcing “noise -> gradient spikes -> policy drift →->more noise” loop and propose StableDRL, a GRPO reformulation that always clips ratios (unconditional clipping) and replaces 1/G normalization with self-normalization by the sum of clipped ratios, plus a staircase attention scheme to enable leakage-free single-pass proxy estimation for block diffusion. Experiments on full-attention and block-diffusion dLLMs show improved stability and state-of-the-art results on multiple reasoning benchmarks.

**Compliance With Llm Reviewing Policy:**

Affirmed.

**Final Justification:**

Due to rebuttal feedbacks from the authors, I maintain my score as weak accept.

**Key Questions For Authors:**

See weaknesses for more questions.

1. Do you apply per-sample gradient norm clipping to ensure the bounded per-sample gradient assumption used in the convex-hull argument? If so, what thresholds and where in the pipeline?
2. Can you report results with multiple random seeds (mean ± std) for key benchmarks and include training traces with confidence intervals? This would strengthen claims of stability and robustness.
3. For fairness, did you re-run ESPO/SPG/WD1 as full-parameter training with matched steps, rollouts, and decoding settings? If not, can you provide matched runs or justify differences (e.g., LoRA versus full-parameter) and their impact?

**Limitations:**

Yes

**Strengths And Weaknesses:**

Strengths:
1. The paper isolates a dLLM-specific instability mechanism centered on noisy, ELBO-based importance ratios and shows why PPO/GRPO’s conditional clipping and fixed group normalization can fail in this regime.
2. StableDRL’s combination of unconditional clipping and self-normalization is a simple, principled modification that targets both individual outliers and group-level magnitude oscillations; the convex-hull interpretation is insightful.
3. The staircase attention mask for block diffusion to perform leakage-free, single-pass ELBO proxy estimation is a practical contribution that appears broadly useful beyond this work.
4. Strong empirical gains across full-attention dLLMs (LLaDA‑8B) and block diffusion (SDAR‑8B), with ablations showing the necessity of both components (clipping and self-normalization).
5. Stabilizing RL for dLLMs addresses a pressing and impactful problem, with results that appear to make full-parameter on-policy fine-tuning feasible where prior methods often required LoRA or early stopping.

Weaknesses:
1. The theoretical results rely on strong assumptions (e.g., single-sample dominance, unbounded noise support) and are presented informally in the main text; while useful for intuition, they do not amount to end-to-end guarantees under realistic training dynamics.
2. Self-normalization introduces bias compared to importance-weighted objectives; the paper argues stability but does not analyze or quantify induced bias or its effect on convergence/optimality.
3. The formulation and notation of the clipping “trust region” are inconsistent: the text states clipping to [1 − \epsilon, 1 + \epsilon] while experiments use ε values as large as 5, 100, 1000; this is invalid if interpreted literally since 1 − \epsilon < 0. The log-space clipping to log(1 − \epsilon) is undefined for \epsilon > 1. This needs correction and a precise definition (e.g., symmetric in log-ratio: \rho \in [e^−\epsilon, e^\epsilon] or \rho \in [1/\epsilon, \epsilon]).
4. The convex-hull bound requires bounded per-sample gradients (B). In practice, B is not guaranteed without explicit per-sample gradient clipping, which is not clearly stated.
5. Baseline fairness is not fully established: some prior methods reportedly use LoRA or early stopping; it is unclear whether re-implementations here are full-parameter with matched training budgets/steps and identical decoding policies. Reports of reward collapse for GRPO/PG could stem from suboptimal settings; a stronger case would include matched compute, identical data, and multiple seeds with error bars.
6.The paper should more directly relate unconditional clipping + self-normalization to established self-normalized/truncated importance sampling and off-policy stabilization (e.g., SNIS, V-trace), and to recent stabilization alternatives for LMs (e.g., logits-convex objectives like LCO, or turn-level/clipping-bias normalization like ST-PPO). Head-to-head empirical comparisons are not strictly required but a deeper conceptual contextualization would strengthen novelty claims.

---

> ### Author Rebuttal · Authors · 2026-03-31
>
> We thank the reviewer for the comments and please find our responses below.
>
> ### W1
> We respectfully argue the main-text theory **does** amount to training dynamic. Table A examines distribution of importance ratio $\rho$ to support the long-tail hypothesis. Across multiple tail-sensitive metrics, estimation noise of $\rho$ is clearly heavy-tailed in the practically small-m regime. For all metrics, higher means more being tong-tailed or exteme valuded.
>
> *Table A*: We perform 8 rollouts on GSM8K, 1000-time MC as oracle ELBO. With bootstrap subset size $m\in\{2,5,10\}$, we measured noise on $\rho$. $\rho_{\max}$, $\rho_{99}$ and $p_{95}$ denote the magnitude of max, 99-th and 95-th percentiles respectively. The dominance ratio, $\max_j|\hat{\mathcal L}_j|/\sum_j|\hat{\mathcal L}_j|$, indicates how a sample dominates estimation of $\rho$.
>
> ||$m=2$|$m=5$|$m=10$|Gaussian|
> |---|---|---|---|---|
> |Excess kurtosis|6.26|2.70|1.34|0|
> |$\Pr\\left(\|\eta\|>2\sigma\right)$|0.072|0.048|0.048|0.046|
> |$\Pr\\left(\|\eta\|>3\sigma\right)$|0.031|0.014|0.010|0.003|
> |$\rho_{\max}$|47.1|11.0|4.5|-|
> |$\rho_{99}$|7.4|3.2|2.2|-|
> |Dominance|0.80|0.57|0.45|$1/m$|
> |Dominance $\left(\|\eta\|>p_{95}\right)$|0.90|0.62|0.42|$1/m$|
>
> Also, We note that Sec. 4.1 is designed to validate how this noise manifests during training. As shown in Fig. 3, (1) for GRPO, increasing spike thresholds and frequencies; (2) for Uncond clip: increasing hitting frequencies.
>
> ### W2
> StableDRL does introduce some bias but significantly reduces variance and improves stability. And we further proves the upper bound on the bias. Let $\bar w_j=\min\{\hat\rho_j,1+\epsilon\}$ be the clipped ratio, $\Delta_r$ denote the difference between the StableDRL update and the standard importance-weighted update on coordinate $r$, and $B_r$ be an upper bound on $|A_j g_j^{(r)}|$. Then
> $$
> |\Delta_r|
> \le
> B_r\,\mathbb E\\left[\left|1-\frac1G\sum_{j=1}^G \bar w_j\right|\right]
> +\frac{B_r}{G}\mathbb E\\left[\sum_{j=1}^G(\hat\rho_j-(1+\epsilon))_+\right].
> $$
> Thus the bias is controlled by two explicit quantities: how far the average clipped weight is from $1$, and how much upper-tail mass is removed by clipping. This is the trade-off made by StableDRL: a controlled bias in exchange for much better stability.
>
> ### W3
> There is a typo in Section 3.5 regarding the clipping of $\epsilon$. In practice, we use a clipping range of $0<\rho<6$. We will explicitly state the upper and lower bounds in the revised version in Sec 3.5.
>
> ### W4, Q1
> We do assume that the per-sample gradient directions are bounded, and will revise accordingly. Empirically, cases where the gradient computation produces NaNs or pathological explosions are extremely rare. In contrast, the reward collapse we observe is more strongly associated with the estimated importance ratio, whose noise can produce large outliers and destabilize training.
>
> On the implementation side, we **do not use** explicit per-sample gradient clipping: computing per-sample gradients would require separate forward-backward passes for each sample in the rollout group, which is prohibitively expensive. We will clarify both the bounded-direction assumption and this implementation detail in the revision.
>
> ### W5, Q3
> On fairness, we provide additional matched reruns. For the full GSM8K table under identical backbone, rollout budget, decoding, optimizer / learning rate / horizon for StableDRL, ESPO, and SPG under both full FT and LoRA; StableDRL is already on par at 200 steps, and the difference is that it remains stable later under full FT.
>
> *Table B*
> |Regime|Method|Acc. @ 200 steps|Acc. @ 1000 steps|Grad norm|
> |---|---|---|---|---|
> |Full FT|StableDRL|82.2|86.2|1.90|
> |Full FT|ESPO| 82.6|N/A|5.00|
> |Full FT|SPG| 82.1|N/A|3.96|
> |LoRA|StableDRL (LoRA)|78.0|84.6|0.32|
> |LoRA|ESPO (LoRA)|79.1|82.3|0.58|
> |LoRA|SPG (LoRA)|78.3|86.1|0.49|
>
> Table B is the same-task Countdown rerun, shows ESPO still exhibits spike-drift-collapse under a fully matched setup.
>
>
> ### W6
> We will add a short positioning discussion and cite these related methods. StableDRL is related in spirit to truncated or self-normalized importance weighting (e.g., SNIS, V-trace), but our setting is different: we study **on-policy** RL for dLLMs, where the importance ratio is itself a **noisy estimated likelihood proxy**. Relative to LM stabilization methods such as LCO and ST-PPO, our method addresses a different mechanism: the dLLM-specific spike-drift collapse induced by noisy proxy ratios.
>
> ### Q2
> Given the rebuttal-time compute budget and the significant amount of resources required to run the full-parameter dLLMs RL, we will add additional seed statistics in the final version if possible.
>
> [1] LCO: Stabilizing Policy Optimization via Logits Convexity, 2026
>
> [2] ST-PPO: Stabilized Off-Policy Proximal Policy Optimization for Multi-Turn Agents, 2025
>
> [3] IMPALA: Scalable Distributed Deep-RL with Importance Weighted Actor-Learner Architectures, ICML 2018

---

> > ### Author Rebuttal · Reviewer_2P3A · 2026-04-03
> >
> > Thank you for the detailed and highly constructive rebuttal. I appreciate the substantial effort put into the new empirical evaluations and theoretical derivations. You have addressed my core concerns effectively:
> >
> > 1.cBias and Theoretical Assumptions: The addition of Table A provides convincing empirical backing for the heavy-tailed noise assumption. Furthermore, deriving the explicit upper bound for the bias introduced by self-normalization thoroughly addresses my concern regarding optimality and formally justifies the bias-variance trade-off.
> >
> > 2. Baseline Fairness: Table B is highly compelling. Demonstrating that StableDRL maintains stability under Full FT while strictly matched baselines (ESPO, SPG) collapse effectively resolves my concerns about baseline fairness and hyperparameter tuning. This strengthens the paper's main claim significantly.
> >
> > 3. Clarifications: Thank you for clarifying the clipping notation typo and the practical limitations regarding per-sample gradient clipping.
> >
> > Please ensure that the corrected notation, the explicit assumptions regarding bounded gradients, the bias upper bound, and the new empirical tables (A and B) are fully integrated into the final manuscript, along with the discussion on related stabilization methods (LCO, ST-PPO, etc.).
> >
> > In light of these comprehensive updates, I keep my score.

---

> > > ### Author Response · Authors · 2026-04-08
> > >
> > > We sincerely appreciate your insightful feedback. Your suggestions will be thoughtfully integrated into the final manuscript, and we thank you for your valuable contribution to improving this work.

---

### Official Review · Reviewer_vJL8 · 2026-03-11

**Soundness:** 3
**Presentation:** 3
**Significance:** 3
**Originality:** 2
**Overall Recommendation:** 4
**Confidence:** 4

**Summary:**

This paper studies why on-policy GRPO become unstable when applied to diffusion LLMs. The core issue is that dLLMs do not provide tractable sequence likelihoods, so the importance ratios must be approximated using Monte Carlo. The authors argue that the resulting noisy ratios estimates interact badly with conditional clipping and group-size normalization. To address this, they propose StableDRL, which replaces conditional clipping with unconditional clipping and replaces normalization by group size with self-normalization by the sum of clipped importance weights. They also extend the approach to block diffusion models via a staircase-attention construction.

**Compliance With Llm Reviewing Policy:**

Affirmed.

**Final Justification:**

The theory and presentation are clearer. And the experimental results are compelling. Figure 2 really shows that the proposed method doesn't collapse like previous methods.

**Key Questions For Authors:**

1. Is the additive noise in (3) a simplification?
2.  What are the assumptions for the statement "The $\exp(\cdot)$ operator maps the symmetric noise $\Delta\eta(x)$ to a long-tailed distribution"?
3. Could the authors argue why the setting of the instability results is realistic in practice?
4. Why are methods trained on different tasks in Fig 3.?
5. Is the method clipping ratios, log-ratios, or only the upper tail?
6. What are the reward functions in each tasks?
7. Why is Clip 1 stopped early in Figure 5 and why results show curves of only 1000 steps?

**Limitations:**

- The method is built around estimated importance ratios, not true importance ratios.
- Also, unconditional clipping and self-normalization clearly reduce variance, but they also induce biais. The paper argues this tradeoff is worthwhile, but does not quantify the bias introduced.
- Stability is only demonstrated over a limited training horizon.
- Generality across rewards and tasks is not established.

**Strengths And Weaknesses:**

## Strengths
- The proposed modification is simple and intuitive: unconditional clipping + self-normalization is easy to understand and straightforward to implement.
- The paper has a solid narrative connecting theory and method. The progression from noisy proxy ratios, to gradient spikes, to policy drift, and then to the proposed fix is easy to follow.
- The benchmark results are strong. On the reported tasks, StableDRL outperforms the listed prior dLLM RL baselines on both LLaDA-8B-Instruct and SDAR-8B.
- The ablations support the claims that both un-clipping and self-normalization are important.


## Weaknesses

**Presentation and Clarity**
- It is not conventional to have a figure above the abstract and because it is a result figure it front-loads benchmark claims before the setting. Figure 2 serves the introduction better.
- The introduction states that “full-parameter GRPO training on dLLMs exhibits an abrupt reward collapse at ~300 steps,” but this is presented very generally. I think that the text should specify which task(s), model(s), and setup this refers to.
- There are several notation and proofreading issues: notation overload around $g$, undefined symbols such as $K$ in 3.4.
- Related work section is centered on recent large masked diffusion models. The presentation would benefit from acknowledging broader seminal work on discrete diffusion like:
```
@inproceedings{austin2021d3pm,
  title={Structured denoising diffusion models in discrete state-spaces},
  author={Austin, Jacob and Johnson, Daniel D and Ho, Jonathan and Tarlow, Daniel and Van Den Berg, Rianne},
  booktitle={Advances in Neural Information Processing Systems},
  year={2021}
}
@inproceedings{campbell2022ctcm,
  title={A continuous time framework for discrete denoising models},
  author={Campbell, Andrew and Benton, Joe and De Bortoli, Valentin and Rainforth, Thomas and Deligiannidis, George and Doucet, Arnaud},
  booktitle={Advances in Neural Information Processing Systems},
  year={2022}
}
@inproceedings{lou2024sedd,
  title={Discrete Diffusion Modeling by Estimating the Ratios of the Data Distribution},
  author={Lou, Aaron and Meng, Chenlin and Ermon, Stefano},
  booktitle={International Conference on Machine Learning},
  year={2024},
}
@inproceedings{shi2024md4,
  title={Simplified and Generalized Masked Diffusion for Discrete Data},
  author={Shi, Jiaxin and Han, Kehang and Wang, Zhe and Doucet, Arnaud and Titsias, Michalis K.},
  booktitle={Advances in Neural Information Processing Systems},
  year={2024},
}
```

**Theory**
- Equation (3) assumes ELBO estimation error as additive noise. Is this a simplification? If yes, it should be described as such.
- In the main text "The $\exp(\cdot)$ operator maps the symmetric noise $\Delta\eta(x)$ to a long-tailed distribution" is only stated informally. I think that assumptions should be made explicit in the main text for that statement.
- The instability results rely on strong assumptions and thus the setting does not seem to be realistic characterization of practical training dynamics. Could the authors argue why the setting is indeed realistic in practice?

**Experiments**
- The footnote of Figure 3 says that GRPO is trained on GSM8K while Unconditional Clipping and StableDRL are trained on Countdown. That makes the figure hard to interpret across methods.
- Figure 4 is also inconsistent: the main text says the ablation is on Countdown, while the caption says GSM8K.
- The clipping needs clarification. The theory describes clipping to $[1-\epsilon,1+\epsilon]$, but experiments use $\epsilon = 5$, then Section 3.5 writes the log space bounds, but $\log(1−\epsilon)$ is undefined is $\epsilon=5$. Is the method clipping ratios, log-ratios, or only the upper tail?
- The paper should be clearer about the reward function used in each experimental setting and about how general the conclusions are across rewards.
- Why is Clip 1 stopped early in Figure 5? Showing longer curves (> 1000 steps) would strengthen the claim.

---

> ### Author Rebuttal · Authors · 2026-03-31
>
> We thank the reviewer for the comments, and please find our responses for Weaknesses on presentation (WP), theory (WT), experiments (WE), Questions (Q), and Limitations (L) below. The new results and clarifications will be added in the revision.
>
>
> ### On Presentation and Clarity
> **WP1:** move Figure 1 to the experiment section;
>
> **WP2:** make the *collapse* statement more precisely by tying it to full-parameter full-attention GRPO-style training on reasoning/puzzle benchmarks;
>
> **WP3:** clean up notations, including removing the overloading of $g$ and defining $K$ (number of blocks) in Sec. 3.4.;
>
> **WP4:** broaden the related work to include discrete-diffusion works such as Austin et al. 2021, Campbell et al. 2022, Lou et al. 2024, and Shi et al. 2024.
>
> **WE2:** The Figure 4 caption is a typo and will be corrected to the Countdown task.
>
> ### On theory
> **WT1,Q1:** In our Eq. (3), the noise $\eta(x)$ stems from **definition** rather than **assumptions or simplifications**. We define the gap between the estimated ELBO and the true value as $\eta(x)$.
>
> **WT2,Q2:** Our assumption is that $\eta(x)$ follows a long-tail distribution after $\exp$. We will provide a formal definition of the long-tail distribution in the revision. We have also verified this assumption experimentally: in practice, with $m=2/5$, the distribution of $\rho(x)$ exhibits significantly high long-tail indicators (see the table below; for details, see the response for *zvqo*)
>
> *Table A*
> ||$m=2$|$m=5$|$m=10$|Gaussian|
> |---|---|---|---|---|
> |Excess kurtosis|6.26|2.7|1.34|0|
> |$\Pr\\left(\|\eta\|>2\sigma\right)$|0.072|0.048|0.048|0.046 |
> |$\Pr\\left(\|\eta\|>3\sigma\right)$|0.031|0.014|0.010|0.003|
> |$\rho_{\max}$|47.1|11.0|4.5|-|
> |$\rho_{99}$|7.4|3.2|2.2|-|
> |Dominance|0.80|0.57|0.45|$1/m$|
> |Dominance $\left(\|\eta\|>p_{95}\right)$|0.90|0.62|0.42|$1/m$|
>
> **WT3,Q3:** We claim that our theory is realistic and consistent with experimental results. We validated the long-tail noise hypothesis in (WT2, Q2). We also validated our theory in Figure 3 of the main text: (1) for GRPO, increasing spike thresholds and frequencies; (2) for Uncond clip: increasing hitting frequencies.
>
> **L1:** The actual importance ratio is unattainable under the dLLMs settings since it involves calculating $n!$ trajectories.
>
> **L2:** StableDRL does introduce bias relative to the standard importance-weighted update, and we will state this explicitly. Let $\bar w_j=\min\{\hat\rho_j,1+\epsilon\}$ be the clipped ratio, let $\Delta_r$ denote the difference between the expected StableDRL update and the expected standard importance-weighted update on coordinate $r$, and let $B_r$ satisfy $|A_j g_j^{(r)}|\le B_r$ for all $j$. Then
> $$
> |\Delta_r|\le
> B_r\,\mathbb E\\left[\left|1-\frac1G\sum_{j=1}^G \bar w_j\right|\right]
> +\frac{B_r}{G}\mathbb E\\left[\sum_{j=1}^G(\hat\rho_j-(1+\epsilon))_+\right].
> $$
> The first term comes from self-normalization, and the second comes from clipping. So the update is biased, but the bias is explicitly controlled; in return, StableDRL avoids the variance spikes that otherwise lead to collapse.
>
> ### On experiments
> **WE1,Q4:** we conducted an experiment both on Countdown to improve the consistency. ESPO still exhibits the same spike-drift-collapse pattern.
>
> |Metric|Pre-collapse (steps 0–90)|Post-collapse (steps 220+)|
> |---|---|---|
> |Mean train reward|1.59|1.08|
> |Peak gradient norm|10.2|24.5|
> |Mean spike rate|0.15|0.30|
>
> **WE3,Q5:** There is a typo in Section 3.5 regarding the clipping of $\epsilon$. In practice, we use a clipping range of $0<\rho<6$. We will explicitly state the upper and lower bounds in the revised version.
>
> **WE4,Q6,L4:** We follow the reward settings provided by ESPO and SPG. Since StableDRL operates on the importance weight estimations, our method is reward agnostic theoretically. We demonstrate task generalization: block dLLMs trained on MATH transfer well to GSM8K and AIME'24. We will discuss the impact of different types of rewards on the stability of DLLM RL in our future work.
>
> **WE5,Q7,L3:** We are resuming training of CLIP1 and will update the Figure 5 in the revision. It does not collapse, but learning proceeds relatively slowly. We limit the training horizon to 1,000 steps because large-scale dLLMs RL is resource-intensive, and 1,000 steps is a widely acknowledged budget. This setting is sufficient to observe the collapse of the baseline and validate our superiority.

---

> > ### Author Rebuttal · Reviewer_vJL8 · 2026-04-04
> >
> > I think that the paper is better now with the added clarifications. I will increase my score to 4.

---

> > > ### Author Response · Authors · 2026-04-08
> > >
> > > We sincerely appreciate your acknowledgment and positive feedback on our work. Your suggestions are invaluable in strengthening the paper, and we will thoughtfully integrate your suggestions into the final manuscript.

---

### Official Review · Reviewer_zvqo · 2026-03-12

**Soundness:** 3
**Presentation:** 3
**Significance:** 3
**Originality:** 3
**Overall Recommendation:** 4
**Confidence:** 2

**Summary:**

This paper investigates the instability issues that arise when Group Relative Policy Optimization (GRPO) is applied to discrete Diffusion Large Language Models (dLLMs). Because exact sequence log-likelihoods are intractable in dLLMs, importance ratios must be estimated (e.g., via ELBO), introducing significant noise. The authors astutely identify a self-reinforcing "instability loop": estimation noise bypasses conditional clipping to produce gradient spikes, inducing policy drift, which in turn amplifies the variance of future ratio estimates and leads to reward collapse.

To break this loop, the authors propose StableDRL, which modifies GRPO with two key changes: (i) unconditional clipping to suppress outlier-induced spikes regardless of advantage sign, and (ii) self-normalization to ensure the update remains bounded within the convex hull of per-sample gradients. The method is also extended to block diffusion models via a novel staircase attention design. Empirical evaluations on reasoning tasks (MATH500, GSM8K, AIME) demonstrate that StableDRL prevents reward collapse and achieves strong performance for both full-attention and block dLLMs.

**Compliance With Llm Reviewing Policy:**

Affirmed.

**Key Questions For Authors:**

1. Can you provide empirical diagnostics from your training runs that validate the key theoretical assumptions, specifically the heavy-tail noise assumption and the frequency of outlier dominance?

2. Does the strict bounding introduced by self-normalization limit the convergence speed of the model during the early stages of training compared to a hypothetical (or low-noise) standard GRPO setup?

3. How sensitive is StableDRL to the choice of the clipping hyperparameter (ϵ)? Furthermore, is the method robust across different proxy estimators or varying numbers of Monte Carlo samples for the ELBO estimation?

4. Is there a critical threshold of ELBO estimation noise where even StableDRL's self-normalization fails to extract a meaningful learning signal, effectively stalling policy improvement?

5. Can you provide tightly matched fairness comparisons against ESPO/SPG under comparable parameter-update or compute budgets?

6. What is the explicit runtime and memory overhead of the staircase attention mechanism compared to iterative leakage-free proxy estimation for block diffusion? An efficiency table would be highly appreciated.

**Limitations:**

yes

**Strengths And Weaknesses:**

Strengths

1. The paper does not merely propose a fix. it rigorously diagnoses the root cause of the failure mode. The decomposition of the estimated ratio into a drift term and a noise term is intuitive, and the formulation of the "instability loop" provides an excellent conceptual anchor.

2. The proposed StableDRL directly and logically addresses the diagnosed issues. The theoretical proof demonstrating how unconditional clipping alone can lead to boundary-saturation, and how self-normalization elegantly resolves this, is a major highlight.

3. The manuscript is exceptionally well-structured. The progression from diagnosis to remedy is clear, and the visual aids concisely communicate the core mechanisms.

4. As dLLMs emerge as a viable alternative to autoregressive models, enabling stable RL fine-tuning is a critical bottleneck. This paper tackles a highly relevant problem with a practical solution.

Weaknesses

1. Some empirical comparisons weaken the causal cleanliness of the argument. For instance, comparing standard GRPO on GSM8K against unconditional clipping and StableDRL on Countdown introduces confounding variables. Furthermore, comparisons against baselines like ESPO/SPG should be explicitly controlled for compute budgets, parameter updates, and adaptation regimes (e.g., LoRA vs. full fine-tuning) to ensure the superiority of StableDRL is purely algorithmic.

2. The theoretical scaffolding relies on strong assumptions (e.g., common tail envelope for noise, outlier dominance). The paper currently lacks empirical diagnostics to validate how often these assumptions hold during actual training.

3. While self-normalization effectively restricts the update to the convex hull of per-sample gradients to prevent spikes, the authors do not discuss the potential trade-off with optimization speed. Strictly confining updates might theoretically limit beneficial large step sizes when the policy legitimately needs to move towards high-reward regions.

4. The transition to block diffusion via "staircase attention" feels slightly disjointed; the paper needs a smoother transition explaining why this extension is a critical proof of generalizability.

---

> ### Author Rebuttal · Authors · 2026-03-31
>
> We thank the reviewer for the comments, and please find our responses on (W)eakness and (Q)uestions below.
>
> ### W1, Q5
> We conducted additional experiments with fair settings to validate our method. （1）For better consistency with Figure 3, we rerun ESPO on **Countdown** under the same setting (backbone, reward, training settings, and full-parameter). ESPO still exhibits the same spike-drift-collapse pattern. (2) We provide a tightly matched 200-step GSM8K comparison across StableDRL/ESPO/SPG under both full FT and LoRA. Although these methods are stable on LORA, they have a lower performance ceiling. In full FT, the baselines still suffer from collapse, validating the effectiveness of StableDRL.
>
> *Table A*
> |Metric|Pre-collapse (steps 0–90)|Post-collapse (steps 220+)|
> |---|---|---|
> |Mean train reward|1.59|1.08|
> |Peak gradient norm|10.2|24.5|
> |Mean spike rate|0.15|0.30|
>
> *Table B*
> |Regime|Method|Acc. @ 200 steps|Acc. @ 1000 steps|Grad norm|
> |---|---|---|---|---|
> |Full FT|StableDRL|82.2|86.2|1.90|
> |Full FT|ESPO|82.6|N/A|5.00|
> |Full FT|SPG|82.1|N/A|3.96|
> |LoRA|StableDRL (LoRA)|78.0|84.6|0.32|
> |LoRA|ESPO (LoRA)|79.1|82.3|0.58|
> |LoRA|SPG (LoRA)|78.3|86.1|0.49|
>
> ### W2, Q1
> We provide an empirical analysis of the distribution of the importance ratio $\rho$ to support the long-tail hypothesis. Across multiple tail-sensitive metrics, the estimation noise of $\rho$ is clearly heavy-tailed in the practically small-$m$ regime. For all metrics, higher values indicate heavier tails or more extreme-valued distributions.
>
> *Table C*
> ||$m=2$|$m=5$|$m=10$|Gaussian|
> |---|---|---|---|---|
> |Excess kurtosis|6.26|2.70|1.34|0|
> |$\Pr\!\left(\|\eta\|>2\sigma\right)$|0.072|0.048|0.048|0.046|
> |$\Pr\!\left(\|\eta\|>3\sigma\right)$|0.031|0.014|0.010|0.003|
> |$\rho_{\max}$|47.1|11.0|4.5|-|
> |$\rho_{99}$|7.4|3.2|2.2|-|
> |Dominance|0.80|0.57|0.45|$1/m$|
> |Dominance $\left(\|\eta\|>p_{95}\right)$|0.90|0.62|0.42|$1/m$|
>
> Here we perform 8 rollouts on GSM8K, 1000-time MC as oracle ELBO. With bootstrap subset size $m\in\{2,5,10\}$, we measured noise on $\rho$. $\rho_{\max}$, $\rho_{99}$ and $\rho_{95}$ denote the magnitue of max, 99th and 95th percentiles respectively. The dominance ratio, $\max_j|\hat{\mathcal L}_j|/\sum_j|\hat{\mathcal L}_j|$, indicates how a sample dominates estimation of $\rho$. We also validated our theory in Figure 3 of the main text: (1) for GRPO, increasing spike thresholds and frequencies; (2) for Uncond clip: increasing hitting frequencies.
>
> ### W3,Q2
> We claim that our method does not impose any restrictions on the gradient step size, but instead adjusts the weights of per-sample gradients. As shown in Table B, our approach did not experience an early slowdown. When applied in an ideal, noise-free setting (such as AR), our method sacrifices optimization speed. However, given the unavoidably high-noise background in dLLMs, our method is crucial for stability.
>
> ### Q3
> For $\epsilon$, Figure 5 shows that $\epsilon\in\{1,5\}$ is stable while $\epsilon\in\{100,1000\}$ collapses. Although increasing the number of MC iterations reduces noise and thereby improves stability, the resulting cost is unacceptable. Our method strikes a balance between efficiency and stability.
>
> ### Q4
> Theoretically, if the noise is extremely high, the learnable signal within that group will be extremely weak. Under these settings, our method may fail to capture the signal, while other methods not only fail to capture the signal but also become unstable due to spikes. Furthermore, we provide a sufficient condition for our method to capture learnable signals ($\langle \hat{u}, u^\star \rangle > 0$): $B\sum_{j=1}^G |\hat{\alpha}_j-\alpha_j^\star| < \|u^\star\|$. Here $\alpha_j^\star$ and $\hat{\alpha}_j$ denote the oracle and implemented normalized clipped weights, and $u^\star, \hat{u}$ are ideal and actual update directions. We will include detailed definitions and proofs in the revised version.
>
> ### W4,Q6
> For block diffusion, we will refine the wording in the revision to enhance readability. In our current naive DDP implementation, staircase attention adds about **15 GB** of memory for **8 groups × 16 rollouts** and takes about **20 s** for MC=2; under the same setting, the iterative leakage-free version has a similar memory footprint but takes about **2 min** because it requires sequential block-wise passes.

---

### Official Review · Reviewer_x3HQ · 2026-03-21

**Soundness:** 3
**Presentation:** 3
**Significance:** 3
**Originality:** 3
**Overall Recommendation:** 4
**Confidence:** 3

**Summary:**

In diffusion models, importance ratios cannot be calculated exactly and must be estimated.
This leads to several problems, gradient spikes and unstable training.
This paper propose StableDRL to mitigate these problem via unconditional clipping and self-normalization.
The authors also provide the theoritical proof that gradient spikes are inevitable once a certain threshold of policy drit is crossed.

**Compliance With Llm Reviewing Policy:**

Affirmed.

**Key Questions For Authors:**

See weakness

**Limitations:**

Y

**Strengths And Weaknesses:**

**Strengths**
- Stable training: the method addresses the root cause of unstable training in dLLMs.
The unconditional clipping ensures that no single sample can derail the entire training process.
- Theory that support method: StableDRL is backed by theoretical proofs. This provides a rigourous guarantee that the gradient norm remains bounded.
- Simple method: The proposed method does not require heavy additional computation or complex auxiliary models.

**Weakness**
- Slower learning rate
- Sensitivity to $epsilon$: the effectiveness of the method depends heavily on the choice of $epsilon$.
- Narrow experiments scope: They conver only language tasks, not molecule or polymer tasks that are common in dLLMs.

---

> ### Author Rebuttal · Authors · 2026-03-31
>
> We thank the reviewer for the comments, and please find our responses below.
>
> ### W1
> We would like to clarify that StableDRL **does not benefit from a smaller learning rate** but instead better **adjusts the weights of the per-sample gradients**. To test this directly, we added a controlled GSM8K-Test comparison across StableDRL, ESPO, and SPG **under the same learning rate** and other settings (backbone, rollout budget, decoding setup, optimizer, etc.), for both full fine-tuning and LoRA.
>
> Table A
> |Regime|Method|Acc. @ 200 steps|Acc. @ 1000 steps|Grad norm|
> |---|---|---|---|---|
> |Full FT|StableDRL|82.2|86.2|1.90|
> |Full FT|ESPO|82.6|N/A|5.00|
> |Full FT|SPG|82.1|N/A|3.96|
> |LoRA|StableDRL (LoRA)|78.0|84.6|0.32|
> |LoRA|ESPO (LoRA)|79.1|82.3|0.58|
> |LoRA|SPG (LoRA)|78.3|86.1|0.49|
>
> These results suggest that StableDRL’s gains do not come from a slower learning rate. At 200 steps, StableDRL is as fast as ESPO and SPG in both settings. The difference emerges during continued full fine-tuning (which has a higher performance ceiling than LoRA), where ESPO and SPG collapse, while StableDRL remains stable and continues to improve, indicating that its gains come from improved training stability rather than slower optimization.
>
> ### W2
> Regarding sensitivity to $\epsilon$, as shown in Fig. 5: StableDRL is stable for $\epsilon\in\{1,5\}$, and $\epsilon=5$ performs better than $\epsilon=1$ because it preserves more valid learning signal while still controlling outliers. In contrast, overly loose thresholds, such as $\epsilon\in\{100,1000\}$, weaken clipping and allow instability to re-emerge, leading to collapse. We will add this interpretation to the revised paper.
>
> ### W3
> Our current paper focuses on the scope: **stable on-policy RL for diffusion language models**. Within this scope, we improve training stability by addressing the issue of high noise in the importance weights and demonstrate strong gains on multiple reasoning benchmarks. Theoretically, the estimation problem associated with the ELBO also exists in discrete diffusion models (e.g., polymer diffusion) in the molecular domain. Therefore, our method is conceptually applicable. However, due to resource constraints, we leave RL in the molecular domain as future work.

---

### Decision · Program_Chairs · 2026-04-30

**Decision:**

Accept (regular)

**Comment:**

This paper analyzes the instabilities that arise when applying GRPO to diffusion LLMs, and proposes a reinforcement learning algorithm called StableDRL that aims to stabilize training of dLLMs by using unconditional gradient clipping and self-normalization. StableDRL is supported by theory proving that the gradient norm remains bounded. The authors also extend StableDRL to blockwise diffusion models using staircase attention. The paper uses StableDRL for RL training over the full set of model parameters (rather than just LoRA parameters), and shows that it outperforms baselines on MATH500 and AIME.

Reviewer x3HQ found that the proposed method is simple, successfully addresses the underlying causes of training instability, and is backed by theory.

Reviewer zvqo found the paper to be well-written with good visualizations, addressing an important problem with a practical solution. The reviewer pointed out that the paper rigorously uncovers the cause of instability and the proposed method addresses the diagnosed issues. The reviewer also noted that the paper provides a proof that helps illustrate the importance of self-normalization.

Reviewer vJL8 found the method to be simple, intuitive, and easy to implement. This reviewer found the paper to be clearly written with a good connection between theory and practice. Regarding the experiments, they noted that the benchmark results are strong and that the ablations are informative.

Reviewer 2P3A found the paper to address an important problem with strong results showing that the stabilization provided by the proposed method makes full-parameter finetuning possible while prior methods needed to use LoRA parameters or early stopping. The reviewer found the method to be simple and principled.

Reviewer x3HQ raised concerns regarding the sensitivity of the proposed method to the $\epsilon$ hyperparameter, the fact that it uses a slower learning rate, and the narrow scope of the experiments which do not include non-language tasks like molecule tasks.

The authors’ rebuttal addressed the learning rate question and provided more discussion of the sensitivity to $\epsilon$. The authors maintain their focus on language tasks as a proof of concept, which I think is reasonable.

Reviewer zvqo raised concerns regarding the strong theoretical assumptions the paper makes, which are not validated to hold in practice. In addition, the reviewer noted that the empirical comparisons are sometimes unclear or not properly budget-matched, and that the paper lacks a discussion of the trade-off between stability and optimization speed.

Reviewer zvqo also raised concerns regarding the sensitivity to the clipping hyperparameter $\epsilon$ and the robustness to different proxy estimators and the number of Monte Carlo samples for ELBO estimation.

The authors addressed most of these concerns in the rebuttal. The authors also provided additional experiments analyzing the distribution of the importance ratios to support the long-tail hypothesis.

Reviewer vJL8 raised concerns regarding the clarity of the exposition, the strong assumptions made in the theory, and the unclear experimental settings and missing experimental details.

Reviewer vJL8 also raised concerns regarding how the proposed method reduces variance but induces bias, and while the paper says the tradeoff is worth it, it does not quantify the bias. In addition, the reviewer points out that stability is only shown over a limited horizon, and that the paper does not show generality across rewards and tasks.

The authors addressed most of these issues in their rebuttal, and the reviewer increased their score to 4.

Reviewer 2P3A raised concerns regarding the strong assumptions made in the theory, the bias introduced by self-normalization, and the fairness of baseline comparisons.

The authors’ rebuttal addressed the reviewer’s concerns.

Overall, the reviewers were unanimous in their support of this paper.  The paper introduces a simple and theoretically-grounded method that improves training stability for diffusion LLMs, and will be of interest to the ICML community.